



# Something fishy going on? Evaluating the Poisson hypothesis for rainfall estimation using intervalometers: results from an experiment in Tanzania

Didier de Villiers[1], Marc Schleiss[2], Marie-Claire ten Veldhuis[1], Rolf Hut[1], and Nick van de Giesen[1]

[1]Department of Water Management, Faculty of Civil Engineering, Delft University of Technology
[2]Department of Geoscience & Remote Sensing, Faculty of Civil Engineering, Delft University of Technology

**Correspondence:** Didier de Villiers (d.j.devilliers@tudelft.nl)

**Abstract.** A new type of rainfall sensor (the intervalometer), which counts the arrival of raindrops at a piezo electric element, is implemented during the Tanzanian monsoon season alongside tipping bucket rain gauges and an impact disdrometer. The aim is to test the validity of the Poisson hypothesis underlying the estimation of rainfall rates using Marshall and Palmer's (1948) exponential raindrop size distribution parameterisation. The latter is defined independently of the scale of observation

and therefore implicitly assumes that rainfall is a homogeneous Poisson process. Our results show that 28.3 % of the total observed rainfall patches can reasonably be considered Poisson-distributed and that the main reasons for Poisson deviations of the remaining 71.7 % are non-compliance with the stationarity criterion (45.9 %), the presence of correlations between drop counts (7.0 %), particularly at higher arrival rates ($\rho_a > 500 \ \mathrm{m}^{-2}.\mathrm{s}^{-1}$) and failing a $\chi^2$ goodness of fit test for a Poisson distribution (17.7 %). Our results show that whilst the Poisson hypothesis is likely not strictly true for rainfall that contributes

most to the total rainfall amount it is quite useful in practice and may hold under certain rainfall conditions. Despite the non-compliance with the Poisson hypothesis, estimates of total rainfall amount over the entire observational period derived from disdrometer drop counts are within 2 % of co-located tipping bucket measurements. Uncorrected intervalometer estimates of total rainfall amount overestimate the co-located tipping bucket measurements by a factor of approximately 3. The overestimate is most likely due to poor calibration of the minimum detectable drop size ($D_{\min}$). Intervalometer estimates of total rainfall

when corrected for minimum drop size are within 1 % of co-located tipping bucket measurements. The intervalometer principle shows good potential for use as a rainfall measurement instrument and for determining rough estimates of mean drop size.

## 1   Introduction

Africa and particularly Sub-Saharan Africa is one of the most vulnerable regions in the world to climate change (Boko et al., 2007). The main economic activity (by share of labour) is agriculture, with 98 % of crop production being rainfed (Abdrabo

et al., 2014). At the same time, much of Sub-Saharan Africa is greatly under-serviced by weather observations and the existing observational networks have been in decline since the mid 1990s; from an average of eight stations per million square kilometres, the density has decreased to less than one in 2015 (data from the Climate Research Unit of the University of East Anglia, 2017). There are some organisations working on setting up new observational networks, such as the Trans-African



Hydro-Meteorological Observatory (TAHMO) but progress is slow due to the lack of financial incentives for weather data

(TAH, 2017). As a result, the African climate has not been as well researched in comparison to Western Europe and the United States (Otto et al., 2015; Washington et al., 2006).

For example, a recent review of weather index insurance for smallholder farmers (some of the world's poorest people) found that the sparsity of ground based weather stations is a large challenge for insurers in Sub-Saharan Africa (Greatrex et al., 2015) and companies have been forced to look to other sources of data or to develop other indices by which to insure crops. Global

rainfall estimates from satellites, such as the Global Precipitation Measurement (GPM) mission are instrumental in bridging this gap. However, satellite observations, whilst providing good spatial coverage, do not cover the entire temporal period and the spatial resolution is often too coarse for local applications. Robust, inexpensive and accurate rainfall measuring instruments would add a lot of value by providing ground based measurements.

Satellite data faces another issue for areas with a lack of ground based data for validation. Since radars do not measure

rainfall directly, information about the raindrop size distribution (DSD) is needed in order to estimate rainfall rates ($R$) from radar reflectivity ($Z$) (Munchak and Tokay, 2008; Guyot et al., 2019). A foundational work in this regard is the exponential DSD model proposed by Marshall and Palmer (1948). Since then, a lot of work has been done on determining alternative parameterisations and many different models have been proposed, of which the most widely used are the exponential, gamma (Ulbrich, 1983; Tokay and Short, 1996; Iguchi et al., 2017) and lognormal distributions (Feingold and Levin, 1986). It has

also been shown that the appropriate parameterisation is dependent on the type of rainfall (Atlas and Ulbrich, 1977) and the climatic setting (Battan, 1973; Bringi et al., 2003). Therefore, ground 'truthing' of DSDs for satellite retrievals is very important to ensure that the natural variability of the DSD is being correctly taken into account when estimating rainfall rates (Munchak and Tokay, 2008).

An assumption that is seldom explicitly mentioned in the presentation of these parameterisations is the homogeneity as-

sumption (Uijlenhoet et al., 1999), which states that below some minimum scale, raindrops are distributed homogeneously (as uniformly as randomness allows) in space and time. Otherwise the parameterisation would depend on the size of the sample volume/area/time period to which it pertains. Statistical homogeneity implies that the number of drops in a fixed volume can be described by a single, constant parameter such as the average drop density per unit volume or the raindrop arrival rate at the surface (Uijlenhoet et al., 1999). Such a point process is called a homogeneous or stationary Poisson point process and

the number of drops is distributed according to a Poisson distribution (Uijlenhoet et al., 1999). The arrival of raindrops at a surface has long been considered an example of a Poisson process (Kostinski and Jameson, 1997; Joss and Waldvogel, 1969). However, this assumption has been questioned and several studies argue that the homogeneity assumption is incompatible with the spatial and temporal clumping of raindrops that is observed in reality. To borrow Jameson and Kostinski's (1997) words; "The 'streakiness' that is part of the lived experience of rainfall can be seen when sheets of rain pass across the pavement

during thunderstorms." This clumping results in greater variability than is predicted by the Poisson hypothesis.

To overcome these difficulties, two different approaches have been proposed. Some researchers like (Lovejoy and Schertzer, 1990; Lavergnat and Golé, 1998) proposed to abandon the Poisson process framework and replace it with a scale dependent, multi-fractal representation of rain. Others, proposed to generalize the homogeneous Poisson process (with a constant mean) to


a doubly stochastic Poisson process or Cox process, where the mean itself is a random variable (Jameson and Kostinski, 1998;
Smith, 1993).

The aim of this study is to formally assess the adequacy of the homogeneous Poisson hypothesis and its importance in deriving rainfall estimates from ground based measurements in a tropical climate. The intervalometer, a new kind of inexpensive rainfall sensor, is introduced and tested for its suitability in providing ground based rainfall estimates in Sub-Saharan Africa. To this end nine intervalometers were deployed over a two month period during the Tanzanian tropical monsoon. Marshall and Palmer (1948) exponential parameterisation of the DSD was used to convert the intervalometer raindrop arrival rates into rainfall rates and results were compared with disdrometer and tipping bucket measurements. A hierarchical system of statistical tests on the drop counts was used to assess the validity of the homogeneous Poisson hypothesis. Section 2 presents the experimental setup. The methods of analysis are detailed in Sect. 3 and the results and discussion are presented in Sect. 4 and Sect. 5 respectively. A list of conclusions follows in Sect. 6.

## 2 Materials

### 2.1 Instruments

In total, the experiment made use of nine intervalometers, one acoustic disdrometer and two tipping bucket rain gauges at eight different sites. The tipping bucket rain gauge was made by Onset (more info at https://www.onsetcomp.com/products/dataloggers/rg3), in the US and was equipped with a HOBO datalogger; the Acoustic Disdrometer was manufactured by Disdrometrics (more info at https://www.disdro.com/) in Delft, The Netherlands; and the Intervalometer was also made by Disdrometrics. The intervalometer is a device that registers the arrival of raindrops at the surface of a piezo electric drum and can be constructed for less than $ 150. It has a minimum detectable drop diameter ($D_{min}$) of 0.8 mm, determined by Pape (2018) in a lab experiment. Typical values of $D_{min}$ for impact disdrometers are between 0.3 mm and 0.6 mm (Johnson et al., 2011). The $D_{min}$ value of 0.8 mm for the intervalometer means that the instrument is likely to miss a lot of small drops and underestimate rainfall rates. The advantage of the intervalometer over a standard rain gauge is that it provides drop counts as well as rainfall estimates. By combining the intervalometer measurements with traditional rain gauge data, a rough estimate of drop sizes can be made. More information about the intervalometer can be found at https://github.com/nvandegiesen/Intervalometer/wiki/Intervalometer or in Pape (2018). The acoustic disdrometer registers the kinetic energy of drop impacts at a drum and converts this to an estimate of the drop size. It is similar to an intervalometer but also provides individual drop size estimates. The minimum detectable drop diameter for the disdrometer is 0.6 mm. A good discussion of the pros and cons of impact disdrometers can be found at e.g. (Tokay et al., 2001; Guyot et al., 2019) and for tipping buckets at e.g. (Ciach, 2003; WMO, 2014). The tipping bucket rain gauge collects all drops over a known surface area and funnels it to a small bucket which tips whenever a fixed volume of water has been collected (0.2 mm). The volume of each tip is verified in situ via a field calibration experiment (FAO, 2001; WMO, 2014).



## 2.2 Experiment

Eight sites were selected along the southern coast of Mafia Island, Tanzania. Figure 1 presents the experimental layout. Sensors were placed in an approximate line, such that a rectangle 3.1 km in length and 500 m in width would cover all the sites. The dimension of the long axis of the experiment was chosen to approximate that of the spatial resolution (approx. 5 km) of the GPM dual polarization radar (DPR) instrument.

**Figure 1.** The eight intervalometer sites on Mafia Island, off the coast of Tanzania. Each site contains one intervalometer. Pole Pole also had a co-located tipping bucket and impact disdrometer. MIL1 also had a co-located tipping bucket.





Rainfall measurement sites were chosen to comply as much as possible with World Meteorological Organisation guidelines within the constraints of accessibility and landscape. Ideally, this means that all of the sensors should be placed in vegetation clearings, sheltered as much as possible from the wind at a height of 1.5 m off the ground and 1.5 m to the nearest instrument (if co-located) and between $2 \times H$ and $4 \times H$ from the nearest object, where $H$ is the difference in height between the nearest obstacle and the rainfall measurement instrument (WMO, 2014). All guidelines were followed except for the $H$ requirement

due to dense vegetation within the entire observational area. In practice, the distance to the nearest object ranged between $H$ and $4 \times H$. No instruments where placed at sites where the nearest obstacle was $\leq H$ away. Tipping buckets were calibrated in the field, in an analogous manner to that described by FAO (2001), by dripping 100 ml of water (from a tripod stand) at a rate slower that 20 mm.h$^{-1}$ onto the instrument and recording the number of tips. The calibration experiment was repeated five times for each tipping bucket to determine the mean volume and the standard deviation (hereafter called std error) of each tip

in the field.

## 2.3  Data Availability

There were some issues over the course of the experiment with the various instruments that affected the availability of data. The disdrometer picked up on a oscillating signal from 20 May 2018 onward that resulted in total corruption of the data. Some intervalometers experienced water damage, particularly in storms with high rainfall intensities, which caused the instruments

to go offline for certain periods of time. Two were damaged beyond repair. The tipping bucket gauges experienced no known issues. Figure 2 presents an overview of the data available.

## 3  Methods

### 3.1  Deriving rainfall rates from rain drop arrival rates

Uijlenhoet and Stricker (1999) present an excellent review of the exponential DSD parameterisation as well as the derivations of

relevant rainfall quantities. A small summary mostly derived from their work is presented below. The raindrop size distribution in a volume of air $N_V(D)$ [mm$^{-1}$.m$^{-3}$] is defined such that the quantity $N_V(D)dD$ represents the average number of drops with diameters between $D$ and $D + dD$ per unit volume of air. Marshall and Palmer (1948) proposed to model $N_V(D)$ using an exponential model of the form:

$$N_V(D) = N_0 \exp(-\Lambda D) \tag{1}$$

$$\text{where} \tag{2}$$

$$\Lambda = 4.1 R^{-0.21} \ [\text{mm}^{-1}] \tag{3}$$

$$N_0 = 8 \times 10^3 \ [\text{mm}^{-1}\text{m}^{-3}] \tag{4}$$

If raindrops are assumed to fall at terminal velocity then $N_V(D)$ can be related to the DSD of drops arriving at a unit surface area, $N_A(D)$ [mm.$^{-1}$.m$^{-2}$.s$^{-1}$], by $v(D)$ [m.s$^{-1}$], which describes the relationship between drop diameter and terminal fall



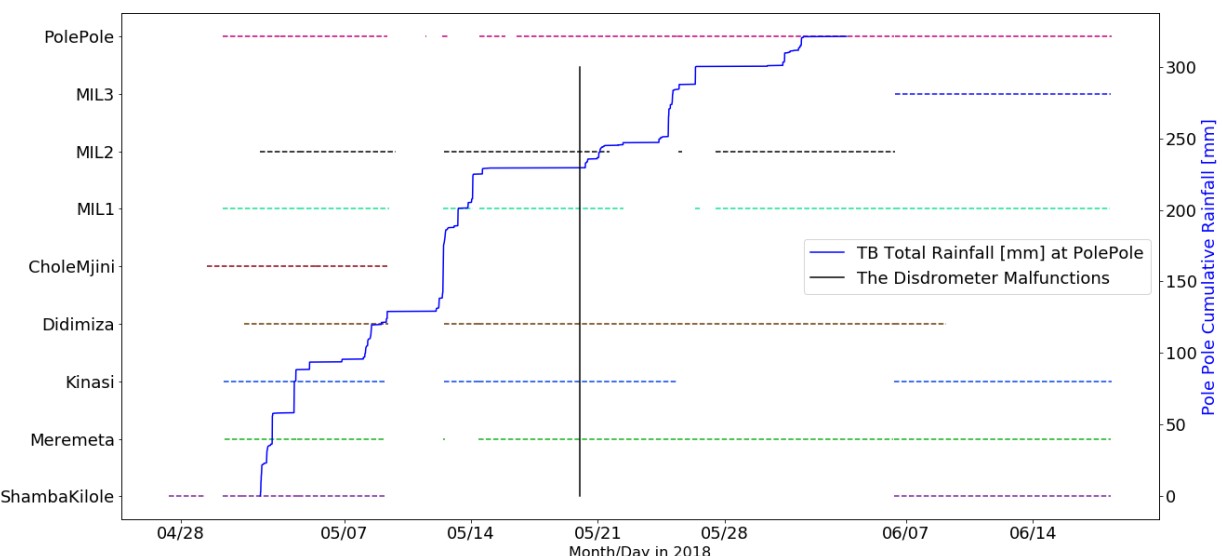

**Figure 2.** Record of the time periods during which the intervalometers collected data for each intervalometer site and the total rainfall amount [mm] from the tipping bucket at Pole Pole.

velocity. $N_A(D)$ is the form of the DSD that is observed by disdrometers and intervalometers (Uijlenhoet and Stricker, 1999; Smith, 1993).

$$N_A(D) = v(D)N_V(D) \tag{5}$$

Atlas and Ulbrich (1977) showed that $v(D)$ can be approximated by a power law, $v(D) = \alpha D^\beta$, with $\alpha = 3.778\,[\mathrm{m.s^{-1}mm^{-\beta}}]$ and $\beta = 0.67\,[-]$ providing a close fit to the data collected on the terminal fall velocity of drops in stagnant air by Gunn and Kinzer (1949) for $0.5\,\mathrm{mm} \geq D \leq 5.0\,\mathrm{mm}$. The mean rainfall arrival rate, $\rho_A\,[\mathrm{m^{-2}.s^{-1}}]$, is defined as the integral over all drop sizes of $N_A(D)$. For the intervalometer this is the integral between $D_{\min} = 0.8$ and $\infty$ since the instrument has a minimum detectable drop diameter of 0.8 mm.

$$\rho_A = \int_{D_{\min}}^{\infty} N_A(D)dD = \alpha N_0 \int_{D_{\min}}^{\infty} D^\beta \exp(-\Lambda D)dD = \alpha N_0 \frac{\Gamma(1+\beta, \Lambda D_{\min})}{\Lambda^{1+\beta}} \tag{6}$$

where $\Gamma$ is the upper incomplete gamma function (Arfken et al., 2013). Uijlenhoet and Stricker (1999) showed that for self consistency purposes, the use of $\Lambda = 4.1R^{-0.21}$ determines that $\alpha = 3.25$, $\beta = 0.762$, which are quite similar to the values presented by Atlas and Ulbrich (1977). Using the Uijlenhoet and Stricker (1999) $\alpha$, $\beta$ values and the Marshall and Palmer (1948) $R - \Lambda$ relationship, the rainfall rate ($R$) can then be calculated from the drop arrival rate ($\rho_A$). The values of $\alpha$, $\beta$, $N_0$



and $D_{\min}$ are fixed and for a given value of $\rho_A$, Eq. 6 is used to numerically solve for $\Lambda(\rho_A|\alpha, \beta, N_0, D_{\min})$. The rainfall rate ($R$) can be estimated by re-arranging the Marshall and Palmer (1948) $\Lambda - R$ relation in Eq. 3 so that $R = \left(\frac{\Lambda}{4.1}\right)^{-4.762}$.

### 3.2 Correcting for biases in the measurement of rainfall rate by the intervalometer

Sources of measurement error for the intervalometer are the calibration of the parameter $D_{\min}$ and the measurement of $\rho_A$. Errors in the determination of $D_{\min}$ affect the $\rho_A - R$ relationship. Errors in the $\rho_A$ measurement can result from splashing of drops from outside the sensor onto the sensor surface during high intensity rainfall (resulting in overestimated rain rates), spurious signals from something other than rain falling on the sensor (resulting in overestimated rain rates), or from edge effects (resulting in underestimated rain rates). Edge effects occur when drops with $D > D_{\min}$ land near the edges of the sensor where the signal is damped and may not be recorded properly, especially if $D$ is close to $D_{\min}$.

There is also model error that arises from the assumption that the DSD is adequately described by the Marshall and Palmer (1948) exponential parameterisation rather than some other parameterisation. Many alternative models for the DSD have been proposed and tested in the literature of which the most widely used are the exponential, gamma (Ulbrich, 1983; Tokay and Short, 1996; Iguchi et al., 2017) and lognormal distributions (Feingold and Levin, 1986). Model error will not be accounted for as the focus of this study is to test the homogeneity assumption that underlies these models rather than compare different DSD models.

If there is additional information about the mean drop size ($\mu_{D_{A,\mathrm{obs}}}$) at different rainfall intensities or arrival rates (for example through an impact disdrometer), an additional constrain can be used to correct for biases in the intervalometer measurement of arrival rate ($\rho_{A,\mathrm{obs}}$) and provide more accurate rainfall estimates. The procedure is as follows: the probability distribution of the drop diameters arriving at a surface per unit time $f_{D_A}(D) = \rho_A^{-1} N_A(D)$ is a gamma distribution (Uijlenhoet and Stricker, 1999); in this case truncated at $D_{\min}$.

$$f_{D_A}(D) = \frac{\Lambda^{1+\beta}}{\Gamma(1+\beta, \Lambda D_{\min})} \times D^\beta \exp(-\Lambda D) \tag{7}$$

$$\beta, \Lambda > 0, \ D \geq D_{\min} \tag{8}$$

The expected value (mean drop size in this case) of a left truncated gamma distribution is given by Eq. 9 (Johnson et al., 2011) and for a complete gamma distribution it is given by Eq. 10 (Uijlenhoet and Stricker, 1999):

$$\mu_{D_{A,\exp}} = E[D_{A,\exp} > D_{\min}] = \left(\frac{1+\beta}{\Lambda_{\exp}}\right) \frac{1 - \frac{\gamma(2+\beta, \Lambda_{\exp} D_{\min})}{\Gamma(2+\beta)}}{1 - \frac{\gamma(1+\beta, \Lambda_{\exp} D_{\min})}{\Gamma(1+\beta)}} \tag{9}$$

$$\mu_{D_{A,\exp}} = E[D_{A,\exp}] = \frac{1+\beta}{\Lambda_{\exp}} \tag{10}$$

Where $\gamma$ is the lower incomplete gamma function (Arfken et al., 2013). All the variables in Eq. 9 are known except for $\Lambda_{\exp}$. Therefore, for a given $\rho_{A,\mathrm{obs}}$ we numerically solve Eq. 6 for $\Lambda_{\exp}$ and the expected mean drop size $\mu_{D_{A,\exp}}$ is calculated from Eq. 9. Now, if the observed mean drop sizes ($\mu_{D_{A,\mathrm{obs}}}$) are some function of $\rho_{A,\mathrm{obs}}$, $f(\rho_{A,\mathrm{obs}})$ these can be incorporated into the parameterisation to constrain the expected mean drop sizes. A good first guess for the form of $f(\rho_{A,\mathrm{obs}})$ is the expectation





of the gamma parameterisation above, but could be any function or simply the observed mean drop size at each rainfall arrival rate.

Now, we can express an expected rainfall rate ($R_{\text{exp}}$) and a 'corrected' rainfall rate ($R_{\text{corr}}$) as functions of the expected and observed mean drop sizes by using the relationship $\Lambda = 4.1\, R^{-0.21}$ and substituting $\Lambda$ in Eq. 9.

$$R_{\text{exp}} = \left( \left( \frac{1+\beta}{\mu_{D_A,\text{exp}} \times 4.1} \right) \frac{1 - \frac{\gamma(2+\beta, \Lambda_{\text{exp}} D_{\min})}{\Gamma(2+\beta)}}{1 - \frac{\gamma(1+\beta, \Lambda_{\text{exp}} D_{\min})}{\Gamma(1+\beta)}} \right)^{\frac{1}{-0.21}} \tag{11}$$

$$R_{\text{corr}} = \left( \left( \frac{1+\beta}{\mu_{D_A,\text{obs}} \times 4.1} \right) \frac{1 - \frac{\gamma(2+\beta, \Lambda_{\text{corr}} D_{\min})}{\Gamma(2+\beta)}}{1 - \frac{\gamma(1+\beta, \Lambda_{\text{corr}} D_{\min})}{\Gamma(1+\beta)}} \right)^{\frac{1}{-0.21}} \tag{12}$$

Divide $R_{\text{corr}}$ by $R_{\text{exp}}$ to get:

$$\frac{R_{\text{corr}}}{R_{\text{exp}}} = \left( \frac{\left[ \frac{1}{\mu_{D_A,\text{obs}}} \frac{1 - \frac{\gamma(2+\beta, \Lambda_{\text{corr}} D_{\min})}{\Gamma(2+\beta)}}{1 - \frac{\gamma(1+\beta, \Lambda_{\text{corr}} D_{\min})}{\Gamma(1+\beta)}} \right]}{\left[ \frac{1}{\mu_{D_A,\text{exp}}} \frac{1 - \frac{\gamma(2+\beta, \Lambda_{\text{exp}} D_{\min})}{\Gamma(2+\beta)}}{1 - \frac{\gamma(1+\beta, \Lambda_{\text{exp}} D_{\min})}{\Gamma(1+\beta)}} \right]} \right)^{\frac{1}{-0.21}} \tag{13}$$

$$\Lambda_{\text{corr}} = 4.1 \times R_{\text{corr}}^{-0.21} \tag{14}$$

The $D_{\min}$ values in the above equations are 0.6 mm for the observed drop sizes (from the disdrometer) and 0.8 mm for the expected drop sizes (from the intervalometer). $R_{\text{exp}}$ is calculated from $\Lambda_{\text{exp}}(\rho_{A,\text{obs}} | \alpha, \beta, N_0, D_{\min})$ as was described at the end of Sect. 3.1. The corrected rainfall rate ($R_{\text{corr}}$) is solved numerically by guessing an initial value for $R_{\text{corr}}$ (e.g. $R_{\text{corr}} = R_{\text{exp}}$),

determining $\Lambda_{\text{corr}}$ from Eq. 14 and iterating until the left and right sides of Eq. 13 are equal. The final value of $R_{\text{corr}}$ is the 'corrected' rainfall rate.

In an analogous manner, if there are additional observations of rainfall rates ($R_{\text{obs}}$), e.g. from a tipping bucket, these can be incorporated to provide unbiased estimates of the mean drop size ($\mu_{D_A,\text{corr}}$) at each rainfall arrival rate. Since a tipping bucket has no minimum detectable drop size, we can combine Eq. 10 with the $R - \Lambda$ relationship to get an expression for $R_{\text{obs}}$:

$$R_{\text{obs}} = \left( \frac{1+\beta}{\mu_{D_A,\text{corr}} \times 4.1} \right)^{\frac{1}{-0.21}} \tag{15}$$

Divide Eq. 15 by Eq. 11 and re-arrange to give an expression for $\mu_{D_A,\text{corr}}$.

$$\mu_{D_A,\text{corr}} = \frac{\mu_{D_A,\text{exp}} \times \left[ 1 - \frac{\gamma(1+\beta, \Lambda_{\text{exp}} D_{\min})}{\Gamma(1+\beta)} \right]}{\left( \frac{R_{\text{obs}}}{R_{\text{exp}}} \right)^{-0.21} \times \left[ 1 - \frac{\gamma(2+\beta, \Lambda_{\text{exp}} D_{\min})}{\Gamma(2+\beta)} \right]} \tag{16}$$

$$\Lambda_{\text{exp}} = 4.1 \times R_{\text{exp}}^{-0.21} \tag{17}$$

This is the same as Eq. 13 re-arranged, except that the expectation of a complete gamma distribution has been used in the

equation for $R_{\text{obs}}$ instead of a truncated gamma distribution. Expected mean drop size ($\mu_{D_A,\text{exp}}$) is calculated from Eq. 9 and $R_{\text{exp}}$ is calculated from $\Lambda_{\text{exp}}(\rho_A | \alpha, \beta, N_0, D_{\min})$. The 'corrected' mean drop sizes can be calculated directly from Eq. 16. Note





that there is a small time delay between the first sensing of a raindrop with the intervalometer and the first tip registered by a tipping bucket. Therefore, in order to compare $R_{\mathrm{obs}}$ and $R_{\mathrm{exp}}$ and derive drop size estimates, rainfall data is aggregated over an entire day.

### 3.3 The Poisson homogeneity hypothesis

The concept of a drop size distribution depends on the assumption that at some minimum spatial or temporal scale (the primary element) the rainfall process is homogeneous. Homogeneity in a statistical sense implies that the data within the primary element follow Poisson statistics (Uijlenhoet and Stricker, 1999). In particular some key assumptions must hold:

1. The rainfall process is stationary, i.e. it has a constant mean raindrop arrival rate.

2. The number of raindrops arriving at the surface over non-overlapping time intervals are statistically independent.

3. The number of raindrops arriving at a surface during a time interval $[t, t + \delta t]$ is proportional to $\delta t$.

4. The probability of more than one raindrop arriving at a fixed surface over a time interval $[t, t + \delta t]$ becomes negligible for $\delta t \to 0$.

Assumptions 3 and 4 are reasonable for small spatial and temporal scales and 1, 2 can be tested. If these fundamental assumptions hold then the distribution of raindrops is given by (eg. (Feller, 2010)).

$$p(k) = \frac{\mu^k \exp(-\mu)}{k!} \tag{18}$$

Where $\mu$ is the average number of drops arriving at a surface per unit time and $k$ is the random number of drops observed during a particular counting interval/window of time. Kostinski and Jameson (1997) show that this simple Poisson model does not explain the clumpiness that is sometimes observed in real rainfall. However, if $\mu$ varies in time and space, then a rainfall event can always be sub-divided into $N$ smaller patches, each of which has its own constant $\mu$. In order to derive the total PDF of the drop counts, it is then necessary to integrate over the probability distribution of the patches $f(\mu)$, resulting in a Poisson mixture.

$$p(k) = \int_0^\infty \frac{\mu^k \exp(-\mu)}{k!} f(\mu) d\mu \tag{19}$$

The variance of the Poisson mixture is greater than the variance of a pure Poisson PDF. Kostinski and Jameson (1997) show that the Poisson mixture provides a better description of the frequency of drop arrivals per unit time than a simple Poisson model. The definition of $f(\mu)$ in Eq. 19 implies that there is a coherence time ($\tau$) over which $\mu$ can be considered stationary and to which the homogeneous Poisson hypothesis can be applied. Therefore, in order to estimate $f(\mu)$ with sufficient accuracy, one requires ($t \ll \tau \ll T$), where $T$ is the entire length of a rainfall event, $\tau$ is the coherence time of a patch and $t$ is the counting interval for the raindrops. Kostinski and Jameson (1997) showed that an order of magnitude difference is sufficient between $t$, $\tau$ and $\tau$, $T$. For the intervalometer data, raindrops are aggregated into 10 second bins. Therefore, the minimum accepted





value for $\tau$ is 100 s and for $T$ it is 1000 s. The length of $\tau$ can be determined by calculating the normalized auto-correlation function for a rainfall event of length $T$ at increasing lag times. The lag time for which the auto-correlation drops below $\frac{1}{e}$ is defined as $\tau$ (Kostinski and Jameson, 1997).

### 3.4 Testing the Poisson homogeneity hypothesis

225  In this study, a rainfall event is defined as a period of rainfall in which the time between consecutive raindrops does not exceed 1 hour. Each rainfall event is sub-divided into $N$ patches of length $\tau$ and the fundamental Poisson assumptions can be tested on each individual patch. A hierarchical test is used, where a patch of rainfall of length $\tau$ must pass each test before moving onto the next test and all tests must be passed in order for a patch to be classified as Poisson. The system of hierarchical tests is as follows.

230  1. **Tests for stationarity:**

   (a) The Augmented Dickey-Fuller (ADF) and Kwiatkowski–Phillips–Schmidt–Shin (KPSS) tests for stationarity are used with a p-value of 0.05. The KPSS test is used to test the null hypothesis that the process is trend stationary (Kwiatkowski et al., 1992). The number of lags considered is equal to $12 \times \left(\frac{n_{\text{obs}}}{100}\right)^{\frac{1}{4}}$ (Schwert, 2012). The ADF test is used to test the null hypothesis that the process has a unit root (Dickey and Fuller, 1979). The lag is determined using the Akaike information criterion (Greene, 2003). The approach to unit root testing implicitly assumes that the time series to be tested can be decomposed into the sum of a linear deterministic trend, a random walk, and a stationary error. The presence of a unit root will result in a trend in the stochastic component and the series will drift away from the deterministic trend value after a perturbation whereas a process without a unit root will not drift after a perturbation. A more complete discussion is presented by (Dickey and Fuller, 1979; Kwiatkowski et al., 1992; Wang et al., 2006). If the null hypothesis for the KPSS test is accepted and the null hypothesis for the ADF test is rejected, then the process is assumed to be strictly stationary (Wang et al., 2006).

   2. **Test for statistical independence:**

   (a) The auto-correlation function of a patch is calculated at increasing lag times. The auto-correlation must be within the 95 % confidence limit (CL) of a Poisson process with $n$ observations (10 s drop counts). If the auto-correlation is zero then the patch auto-correlation is known to be approximately normally distributed with mean $\mu = \frac{-1}{(n_{\text{obs}}-1)}$ and variance $\sigma^2 = \frac{(n_{\text{obs}}-2)}{(n_{\text{obs}}-1)^2}$, provided the number of observations ($n_{\text{obs}}$) from which the auto-correlation is calculated is large in comparison to the number of time lags considered (Haan, 1977) and the largest time lag is greater than $\frac{\tau}{5}$ (Maity, 2018). The criterion $\frac{\tau}{4}$ is used in this study. The 95 % confidence limits for the auto-correlation function have been defined as $\mu \pm 1.96\sigma$ (Uijlenhoet and Stricker, 1999).

250  3. **Test for goodness of fit:**

   (a) A one-way $\chi^2$ test (Pearson, 1900) for the goodness of fit between the observed frequencies and the expected frequencies of a Poisson distribution with the same mean is conducted. A p-value of 0.05 is used.





4. **Disperion criterion quality check**

    (a) Dispersion is defined as the the ratio of the patch variance to the patch mean. According to Hosking and Stow (1987), the dispersion index calculated from a random rainfall patch of $n$ observations drawn from a Poisson distribution has mean $\mu = 1$ and standard deviation $\sigma = \left(\frac{2}{(n_{\mathrm{obs}}-1)}\right)^{\frac{1}{2}}$. Like for the auto-correlation function, $\mu \pm 1.96\sigma$ has been defined as the 95 % confidence limits for the Poisson dispersion index.

5. **Sample independent quality check**

    (a) Kullback (1968) (KL) divergence is also known as the relative entropy between two probability density functions. Here, the KL divergence is calculated to give an indication of how well the observed distribution matches the Poisson distribution (independently of sample size) (Hershey and Olsen, 2007). A value of zero for the KL divergence indicates that the two distributions in question are identical.

Tests 1 and 2 assess the stationarity and independence assumptions of a Poisson process. Test 3 checks that the distribution matches a Poisson distribution and Tests 4 and 5 are quality checks. The quality checks are used because the sample size over which each test is conducted is often quite small. Figure 3 shows a good example of a patch of rainfall that passes all of the tests and can therefore reasonably be assumed to comply with the Poisson homogeneity hypothesis.

The rainfall rate is plotted in the top panel and can be characterised by uncorrelated fluctuations around a constant mean rate of arrival, in this case $220\,\mathrm{m}^{-2}.\mathrm{s}^{-1}$. The corresponding probability density function (pdf) of this patch of rainfall along with the expected pdf function of a Poisson process with the same mean arrival rate is plotted in the bottom panel. The auto-correlation function of the patch is plotted in the middle panel.

## 4 Results

### 4.1 Rainfall Rates

The total rainfall amounts [mm] measured by the co-located tipping bucket, intervalometer and disdrometer at the main site (Pole Pole) for the longest 'online' period of the three instruments are presented in Fig. 4. Estimates of total rainfall are derived from the arrival rates using the Marshall and Palmer (1948) exponential parameterisation for both the disdrometer and intervalometer. The disdrometer estimate is within 2 % of the tipping bucket record. However, the intervalometer overestimates the total rainfall compared to the tipping bucket by a factor of more than 3. Figure 4 also shows that the intervalometer recorded much higher arrival rates than the disdrometer over all rainfall events despite having a smaller sensor area and a larger minimum detectable drop size. After correction of the intervalometer rainfall estimates (using Eq. 13) by the observed mean drop diameters (from the impact disdrometer), results were in good agreement (within 4 %) with the tipping bucket record as a whole.

In Fig. 5 the performance of the corrected parameterisation for intervalometer data over three rainfall periods is presented for Pole Pole (left side) and MIL1 (right side). In panel (c), the corrected parameterisation provides a good estimate of within

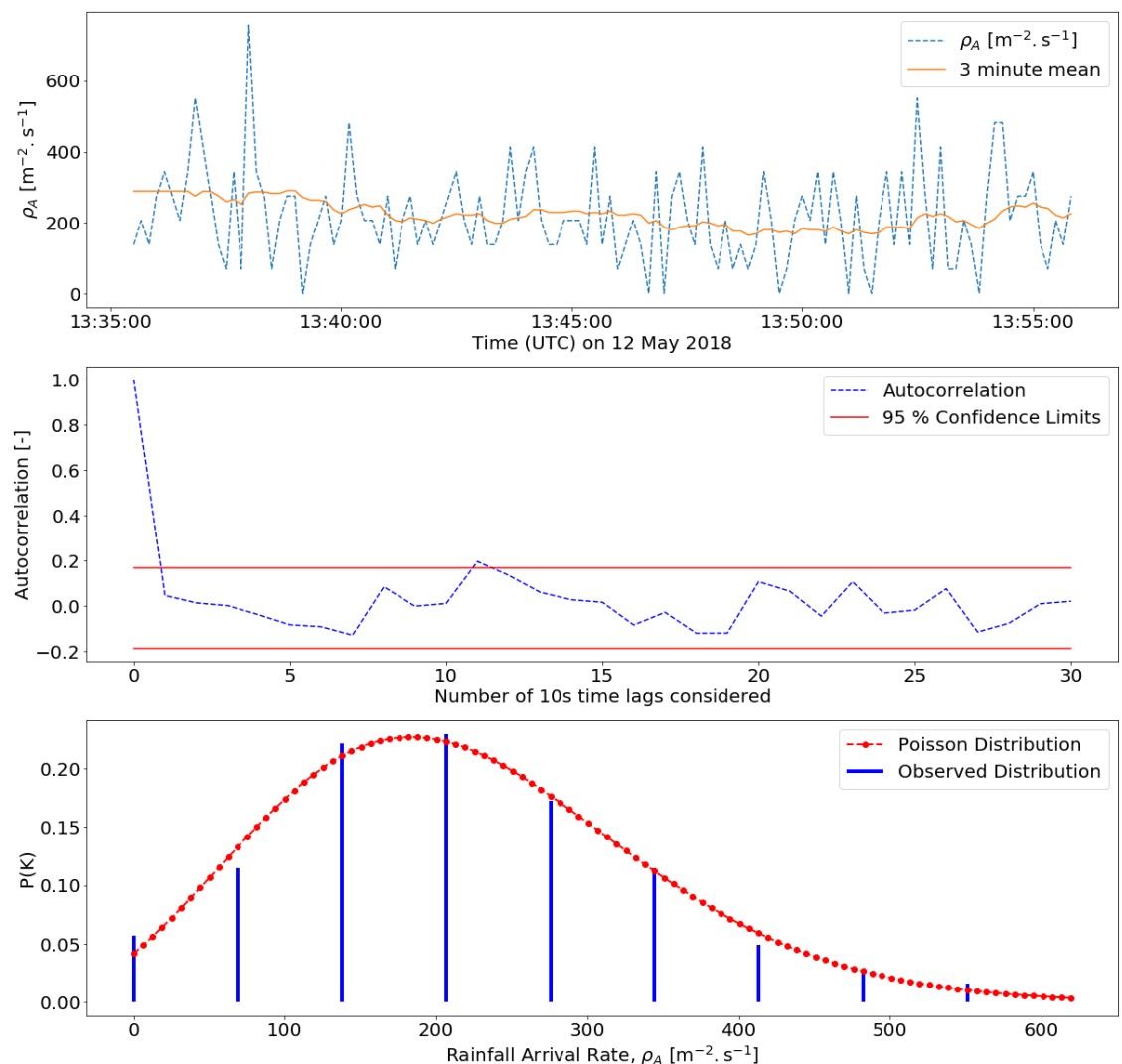

**Figure 3.** A patch of rainfall, with a coherence time of 20 minutes, that can reasonably be assumed to be an example of a Poisson process. The dispersion of the patch is 1.1 and the KL-divergence is 0.01, indicating very good agreement between the observed pdf of the patch and the expected pdf from Poisson.

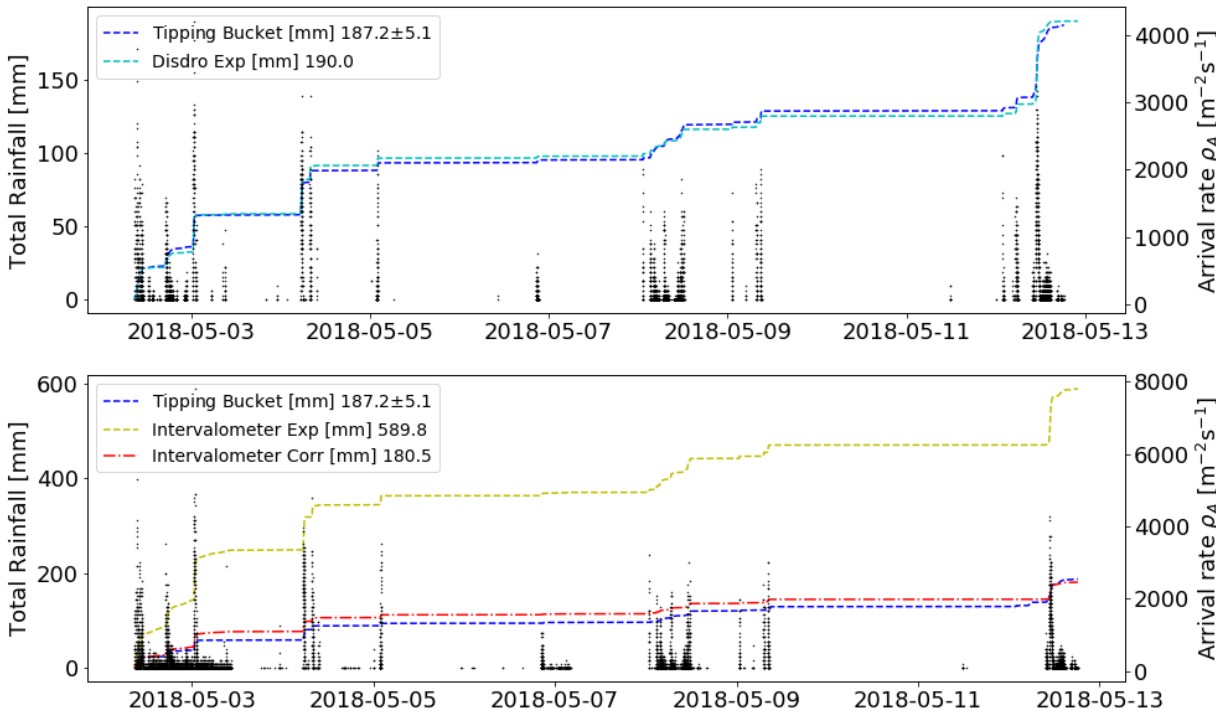

**Figure 4.** The total rainfall amount [mm] observed by the co-located tipping bucket, intervalometer and disdrometer at the main site (Pole Pole) for the longest 'online' period of the three instruments. The suffixes 'Exp' and 'Corr' refer to the uncorrected rainfall from the exponential parameterisation and the corrected rainfall, respectively. Also plotted are the rainfall arrival rates measured by the disdrometer and intervalometer.

4 % of the tipping bucket value. For panels (a) and (e) the agreement of the corrected estimates is poorer. In panel (a), the intervalometer overestimates by 16 % and in panel (e), it underestimates by 26 % compared with the tipping bucket value. The rainfall periods in panels (a) and (e) contain patches of sustained low intensity rainfall. This is not the case for panel (c), where the rainfall occurs mostly in short intense bursts typical of convective rainfall. The intervalometer correction derived from the main site was also applied to the MIL1 site measurements (right hand side of Fig. 5), where tipping bucket measurements are also available. Note that MIL1 is situated approximately 1 km from Pole Pole. The drop sizes used to perform the correction might therefore not be optimal for MIL1. Nonetheless, the correction gives good estimates at this site in two out of the three rainfall events. In panels (b) and (d) for MIL1 the corrected parameterisation provides good estimates of the tipping bucket values, within 4 % and 3 % respectively. In panel (f) the corrected rainfall rates are underestimated by 22 %.



**Figure 5.** The exponential parameterisation is used to estimate rainfall rates for three different rainfall events at Pole Pole (left three panels) and MIL1 (right three panels). The corrected rainfall (Corr) is compared to the tipping bucket (TB) measurements as well as the un-corrected measurements (Exp). The rainfall arrival rates are also plotted.





The total cumulative rainfall estimates over the entire data record of the intervalometers at Pole Pole and MIL1 are presented in Table 1. The presented values are a cumulative sum over each of the individual rainfall events within the rainfall record. The
295 table shows that the corrected intervalometer estimates are in excellent agreement with the tipping bucket values for Pole Pole (within 1 %) and in reasonable agreement with the tipping bucket measurements at MIL1 (within 13 %). Note that MIL1 is located more than 1 km from where the drop sizes were observed.

**Table 1.** Total cumulative rainfall [mm] over the whole measurement period of the intervalometers compared to the total rainfall amount [mm] for the tipping buckets at MIL1 and Pole Pole.

| Instrument and Parameterisation | Pole Pole (Main Site) | MIL1 |
|---|---|---|
| Intervalometer, Uncorrected | 814.4 | 108.6 |
| Intervalometer, Corrected | 249.3 | 33.2 |
| Tipping Bucket | $251.0 \pm 6.8$ | $37.9 \pm 1.1$ |

The intervalometer estimates and co-located tipping bucket measurements are also used to derive estimates of the mean drop sizes, via Eq. 16. The estimated drop sizes are plotted alongside the disdrometer observed mean drop sizes as well as the
300 expected values from the parameterisation in Fig. 6. The drop size estimates are derived from 17 data points corresponding to 17 days of measurements. This is because tipping bucket and intervalometer measurements were aggregated into daily averages to account for the time delay between sensing a drop and the first tip. Days with less than 2 mm of total rainfall were discarded and not used to determine the mean drop sizes to mitigate the effect of evaporation and spurious drop counts which can be significant at low rainfall rates. The estimated drop sizes roughly match with the observed values and are within one standard
305 deviation of the observed mean drop sizes at all considered rainfall arrival rates. However, the observed standard deviation of the drop sizes is quite high and of a similar size to the total change in the mean drop size over the range of arrival rates.

### 4.2 Testing the Poisson Hypothesis

The coherence time or window length over which the Poisson tests were performed ranged from 2 to 22 minutes across all eight sites, with a typical length being in the order of 6 minutes. Using the tests defined in Sect. 3.4, we determined the rainfall
310 patches that can reasonably be assumed to be representative of a Poisson process.

The proportion of rainfall patches, averaged across all the intervalometers, that do not conform with the Poisson hypothesis as well as the mean arrival rate for each group is presented in Fig. 7. Overall, only 28.3 % of all patches can reasonably be assumed to be Poisson distributed. These are patches of stationary rainfall that exhibit no correlation between drop counts within a 95 % confidence interval, match a Poisson distribution very well and have a mean dispersion of approximately 1.
315 The KL divergence of the Poisson patches was between 0.01 and 0.07 for all sites and only between 0 % and 7 % of all those patches had a KL divergence greater than 0.2.



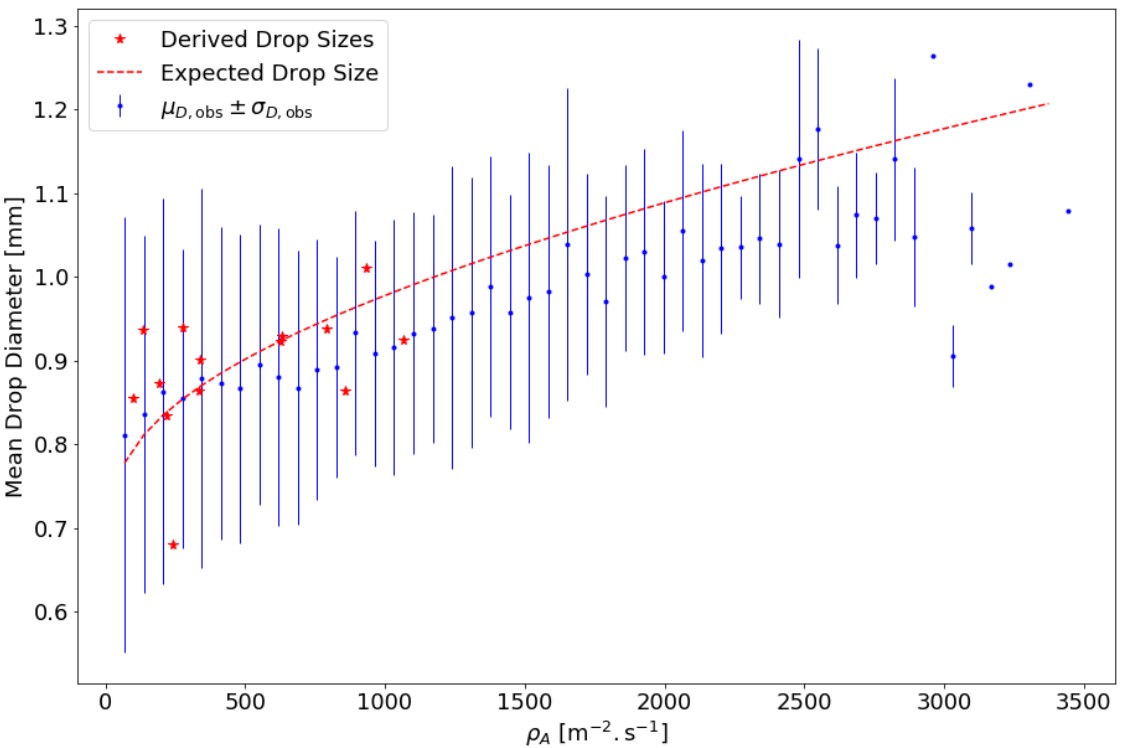

**Figure 6.** Estimated mean drop sizes, derived from Eq. 16 using a combination of intervalometer estimates and tipping bucket measurements of total rainfall amount, versus raindrop arrival rates. The derived estimates, disdrometer observed mean drop sizes and expected mean drop sizes (from the exponential parameterisation) are plotted.

45.9 % of all patches failed the stationary tests and 7.0 % did not pass the independence test, indicating the presence of correlations between drop counts on scales as small as 2 minutes. It should be noted that these patches of rainfall are characterised by higher arrival rates (e.g. the rainfall that fails the independence test has a mean $\rho_A$ that is approximately 4 times higher than the $\rho_A$ of Poisson rain).

Of the remaining 47.1 % of rainfall patches, 17.7 % did not follow a Poisson distribution. Only a very small subset (1.2 %) did not pass the dispersion criteria and mostly because the observed variance was larger than expected for Poisson statistics. Again, these patches were characterised by higher raindrop arrival rates than the ones that passed.

Based on the previous results, it appears that most rainfall patches with higher raindrop arrival rates are inconsistent with the Poisson hypothesis. This can be clearly seen in the two middle panels (c, d) of Fig. 8 as well as panel (b). The time series in Fig. 8 clearly show that the mean rainfall arrival rate is a reasonable predictor of whether a given patch is likely to be Poisson or



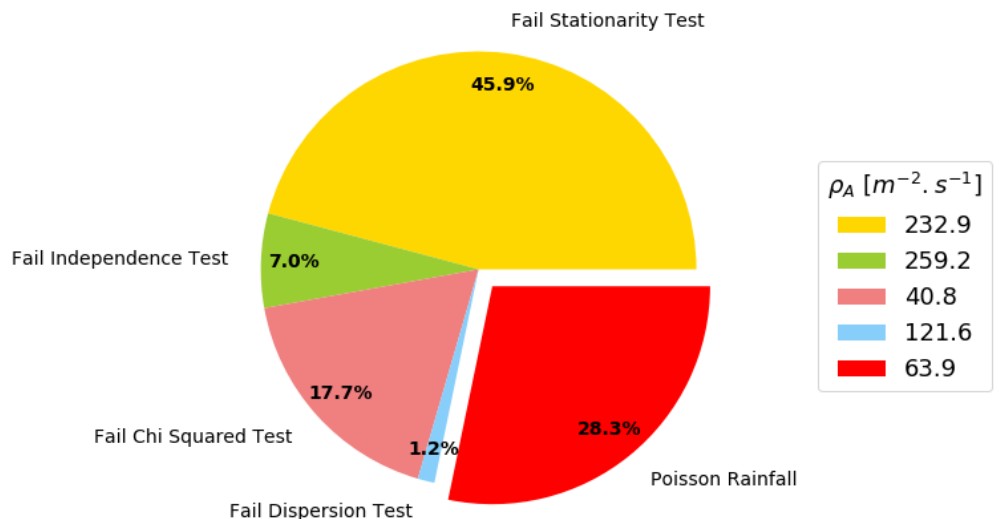

**Figure 7.** The percentage of all rainfall patches, measured by the intervalometer, that fail each of the hierarchical tests as well as the mean rainfall arrival rate for each group. The presented data is an average, weighted by the length of each patch, across all of the intervalometer sites.

not. Figure 8 shows the total rainfall record for Pole Pole, Chole Mjini and Meremeta in the left hand column and a single large scale storm that was observed at all three sites in the right hand column. This storm is characterised by sustained stratiform type rainfall with low arrival rates and little fluctuation over time. This type of rainfall pattern is quite atypical for the rainfall record as a whole. Chole Mjini was only online for a relatively short period of time between 30 April 2018 and 08 May 2018 and this period happened to contain this atypical storm. The much longer time series for Pole Pole (panel a) and Meremeta (panel e) show that the observational record is dominated by intermittent rain events with sharp peaks and lots of convective rainfall followed by longer dry spells. Figure 8 also shows that most rainfall patches and in particular patches of rain with high rainfall arrival rates are typically not classified as Poisson, whereas many patches of rainfall with sustained low arrival rates (below $500\,\mathrm{m}^{-2}.\mathrm{s}^{-1}$) are classified as Poisson. This is especially evident in panels (b) and (d) where the two rainfall peaks do not pass the Poisson tests but the lower intensity patches in between them do.

The disdrometer drop size measurements can be used to characterise Poisson and non-Poisson rainfall patches further and are presented in Fig. 9. The mean drop size of each of the 10 s drop counts is plotted in panels (a) and (b) whereas panel (c) contains the average drop size of all the 10 s drop counts at each arrival rate. The larger variance in mean drop size at lower arrival rates is due to the fact that these 10 s drop counts contain fewer drops and therefore the mean is more susceptible to random sampling effects. The trend in mean drop size with rainfall arrival rate for Poisson and non-Poisson rain is presented





**Figure 8.** The occurrence of Poisson rain in the rainfall records of three sites. The complete observational period is plotted in the left hand column and a single large scale storm which was common to all three sites is plotted in the right hand column. The observational period of Meremeta and Pole Pole is much longer than Chole Mjini due to an instrument failure at that site.





in the top panel. It shows again that Poisson rain is characterised by low arrival rates. No examples of Poisson rain are found at $\rho_A > 1500 \, \mathrm{m^{-2}.s^{-1}}$. The data also shows a positive correlation between the mean drop sizes and the arrival rate as is expected by the Marshall and Palmer (1948) parameterisation.

The middle panel of Fig. 9 presents, for each data point, the test that it fails. It shows that Poisson rain is found mostly at the lower end of the arrival rate range, $\rho_A \leq 500 \, \mathrm{m^{-2}.s^{-1}}$. This range of rainfall arrival rates contributes little to the total rainfall, 69 % of all drops fall in this range but only contribute 16 % to total rainfall. At arrival rates between 500 and 1300 $\mathrm{m^{-2}.s^{-1}}$ the rainfall is a mixture of Poisson rain and mostly patches of rainfall that fail the $\chi^2$ test. Data that fail the $\chi^2$ test are patches of stationary rainfall with uncorrelated fluctuations about the mean. However the data are over or under dispersed

compared to the expected Poisson value of 1 and do not match the Poisson distribution. Mostly, this data is over-dispersed, i.e. the variance is greater than expected by Poisson statistics. As arrival rate increases to between 1300 and 2000 $\mathrm{m^{-2}.s^{-1}}$, a higher proportion of rainfall (in this sub-range) fails the stationarity and independence tests indicating that rainfall is becoming more and more dynamic (rapid changes in the mean and correlations between drop counts). At arrival rates greater than 2000 $\mathrm{m^{-2}.s^{-1}}$ the patches of rainfall predominantly fail the stationarity test. Arrival rates greater than 1000 $\mathrm{m^{-2}.s^{-1}}$ systematically

fail the independence tests and arrival rates greater than 2000 $\mathrm{m^{-2}.s^{-1}}$ systematically fail the stationarity tests. This rainfall is characterised by correlations between drop counts and fluctuations in the mean arrival rate on scales of 2 to 22 minutes.

In the bottom panel trends in the mean drop size for Poisson and non-Poisson rain are presented, as measured by the disdrometer. The expected mean drop size of the parameterisation at each arrival rate is also shown. The expected drop sizes are slightly overestimated compared to the observed drop sizes, although they are well within one standard deviation for most

arrival rates. The overall agreement between the expected and observed drop sizes is quite good over the range of arrival rates between 500 and 2500 $\mathrm{m^{-2}.s^{-1}}$, which contributes 63 % to the total rainfall amount. The parameterisation overestimates most of the drop sizes at arrival rates greater than 2500 $\mathrm{m^{-2}.s^{-1}}$, where the data become quite sparse. The positive trend in mean drop size predicted by the parameterisation is not as clear for the Poisson data as for the non-Poisson data. At arrival rates less than 700 $\mathrm{m^{-2}.s^{-1}}$ the Poisson mean drop sizes are larger than the parameterisation and non-Poisson values and at arrival rates

greater than 700 $\mathrm{m^{-2}.s^{-1}}$ the opposite is the case. However, at arrival rates greater than 700 $\mathrm{m^{-2}.s^{-1}}$ the Poisson data become quite sparse and overall the data fall within the range of the large standard deviation.



**Figure 9.** Trends in mean drop size for Poisson and non-Poisson rain are presented as well as the percentage of drops that fail each of the Poisson tests. The top panel differentiates between Poisson and non-Poisson rain. The middle panel is further subdivided to show which of the Poisson tests each data point fails. The bottom panel shows the observed mean drop sizes with error bars of one standard deviation for Poisson and non-Poisson drops as well as the parameterised values.





## 5 Discussion

### 5.1 Rainfall Rates

Accurate estimates of total rainfall can be derived using Marshall and Palmer (1948)'s parameterisation from disdrometer
arrival rate measurements. This is because the expected mean drop size of the parameterisation shows good agreement with
the observed mean drop sizes. In particular, the expected and observed values match reasonably closely over the range of
rainfall arrival rates that contribute most to the total rainfall (63 % of total rainfall occurs between 500 to 2500 $\mathrm{m}^{-2}.\mathrm{s}^{-1}$).
The parameterisation under-estimates the observed mean drop size at low arrival rates $\rho_A \leq 500$ in comparison to observed
values. However, it is known that impact disdrometers underestimate the number of small drops and the number of drops in
general due to the truncation of drops below the detection limit. Therefore, the difference between the parameterisation and the
observed values could be a result of under-reporting of small drops by the instrument that leads to underestimated rainfall rates
and over estimated mean drop sizes at low arrival rates. The parameterisation overestimates the mean size of drops at arrival
rates greater than 2500 $\mathrm{m}^{-2}.\mathrm{s}^{-1}$ which leads to over-estimated rainfall rates at high drop arrival rates.

For the intervalometer, the situation is different. Total rainfall amounts are over-estimated by a factor of approximately 3
compared with tipping bucket rain gauges. This is because the intervalometer registers higher drop arrival rates during each
rainfall event compared to the disdrometer. This is surprising as the intervalometer has a smaller measurement area and a
larger $D_{\mathrm{min}}$ value than the disdrometer. The possible reasons for the overestimation are splashing from the intervalometer
housing onto the sensor during intense rainfall events, spurious drops due to electromagnetic or physical interferences or the
fact that the minimum detectable drop diameter may actually be smaller than 0.8 mm. Comparison of the rainfall arrival
rate records for the disdrometer and intervalometer, for example in Fig. 4, show that when the intervalometer senses rain,
so does the disdrometer and vice versa. This excludes overestimation by an interfering signal which would also be expected
to register outside the rainfall periods. The intervalometer overestimates are not constrained to intense rainfall periods but
occur throughout the rainfall record. These two findings indicate that spurious drops and splashing are unlikely causes for the
higher arrival rates registered by the intervalometer. It is most likely that the parameter $D_{\mathrm{min}}$ was poorly determined and the
intervalometer registers drops that are smaller than 0.8 mm.

The value of $D_{\mathrm{min}}$ can be calculated by numerically determining which value of $D_{\mathrm{min}}$ results in the closest match between
the intervalometer estimate of total rainfall and the tipping bucket measurement. The intervalometer $D_{\mathrm{min}}$ value is 0.47 mm
when determined in this way which is close to the disdrometer value. Therefore, it is reasonable to assume that the drop
sizes observed by the intervalometer are of a similar size to those observed by the disdrometer. Using this assumption the
intervalometer results are corrected by incorporating the disdrometer observed values of the mean drop size. This results in
accurate rainfall rates for the intervalometer compared to the tipping bucket (within 1 %) for the entire experiment.

The correction results in fair estimates at another intervalometer site approximately 1 km away (within 13 % of the tipping
bucket value). This indicates that the observed mean drop size and therefore the DSD is reasonably stable over spatial scales
of 1 km. The correction also gives good results outside the period of time when the disdrometer was online. The last rainfall
estimate from the intervalometer is approximately 1 month later than the last measurement by the disdrometer. This indicates





that the average DSD is also reasonably stable over the entire two month period of the experiment. The derivation of reasonably accurate rainfall measurements with both the disdrometer and the intervalometer indicates that Marshall and Palmer's (1948) parameterisation of the DSD is a good approximation of the observed DSD over the period of the experiment. These results show proof of concept for the intervalometer as a ground based rainfall measuring instrument.

The intervalometer also shows good potential for being used to derive rough estimates of the mean drop size as a function of rainfall rate. Using only 17 data points it was possible to estimate mean drop sizes that were within one standard deviation of the disdrometer observed mean drop sizes. However, more work, with a larger data-set is necessary to fully assess the validity of using intervalometer measurements combined with rainfall gauge amounts for deriving estimates of mean drop size.

## 5.2  Testing the Poisson Hypothesis

The results of the hierarchical tests show that the majority of rainfall tested does not comply with the Poisson homogeneity hypothesis. This is because the rainfall record is dominated by dynamic convective storms that are characterised by high arrival rates that are fluctuating on very short time scales ($< 2$ min in some cases). This rainfall is also characterised by correlations between drop counts on these time scales. This convective type rainfall that contributes most to the total rainfall amount ($>$ 80 %) in this study is almost never classified as Poisson and does not exhibit characteristics that are consistent with Poisson
statistics.

Another type of rainfall is also observed in the rainfall record. This stratiform type rain is characterised by sustained periods of consistent low intensity rainfall that has few fluctuations in the mean arrival rate. Rainfall of this type is often classified as Poisson and appears to exhibit characteristics that are consistent with Poisson statistics yet it contributes less than a fifth to the total rainfall amount. Furthermore, rainfall estimates derived from the Marshall and Palmer (1948) model over rainfall events
with sustained low intensity rainfall are worse than the rainfall estimates for dynamic convective storms when compared to co-located tipping bucket values.

How do we explain the fact that rainfall estimates based on a parameterisation which has been defined independently of a notion of scale and therefore implying homogeneity, are excellent for both disdrometer arrival rates and corrected intervalometer arrival rates? At the same time the majority of rainfall does not comply with the Poisson hypothesis and the presence of
rainfall that is often classified as Poisson appears to result in worse rainfall estimates. Is something fishy going on?

The regime of tests implemented in this study aim to assess the validity of the Poisson hypothesis in rainfall estimation. I.e. the tests are binary (yes vs no) in nature. We find that for most of the rainfall the Poisson hypothesis is not strictly true. However, the usefulness of the Poisson hypothesis is not tested. This approach may be too short-sighted and other, more practically oriented diagnostic tools could be designed to determine the conditions under which the Poisson hypothesis is
likely to result in good estimates of rainfall rates (or drop diameters). So, whilst the Poisson model may not be strictly true for the rainfall observed in this study it does appear to be a good approximation and highly useful for estimating rainfall rates.

The worse quality of rainfall estimates at low arrival rates is likely due to the sampling uncertainty of the intervalometer. It is known that impact disdrometers and the intervalometer underestimate the numbers of small drops due to the minimum detectable drop size and therefore also the rainfall rate at low rainfall arrival rates. At low arrival rates measurements are also





more susceptible to spurious drop counts. The worse estimates of rainfall at low arrival rates is likely due to these measurement effects rather than whether they are Poisson or not.

There is also the issue that the regime of tests used in this study is likely biased such that rainfall with lower arrival rates is much more likely to be classified as Poisson than rainfall with higher arrival rates. This is due to inherent differences between low and high rainfall arrival rates and also the failure of the $\chi^2$ goodness of fit test to reject the null hypothesis at small sample

sizes. The majority of low arrival rate rainfall generally occurs in patches of rainfall characterised by reasonably stationary mean arrival rate and uncorrelated fluctuations around this mean. High arrival rates occur in highly dynamic patches of rainfall that have changes in the mean at smaller time scales than most of the patches tested in this study. Consequently, almost no rainfall with high arrival rates passes the stationarity and independence tests whereas a very large proportion of rainfall with low arrival rates does. The $\chi^2$ goodness of fit test is then conducted almost exclusively on patches of rainfall with low arrival

rates. These patches have small sample sizes and the power of the $\chi^2$ test to reject the null hypothesis is limited at these sample sizes.

This is well understood in statistics and has led to various sampling criteria such as a minimum of five observations per rainfall arrival rate class for the $\chi^2$ goodness of fit test (Conover, 1999). This criterion is not used in this study. However as pointed out by Kostinski and Jameson (1997); Jameson and Kostinski (1998), rainfall conditions are changing rapidly,

sometimes on temporal scales smaller than 2 minutes. The presence of these fine structures within rainfall would be obscured by larger sampling windows. Furthermore sampling across such structures with different means may actually lead to increased uncertainty in the mean. Similarly, the auto-correlation can no longer be calculated making it hard to define patches on which the Poisson assumptions can be tested. This increased uncertainty in the mean over an entire rainfall event would make it almost impossible to test the homogeneous Poisson hypothesis because rainfall is very rarely stationary over longer time periods.

The high acceptance rate of the Poisson hypothesis at low arrival rates observed in this study may be driven by the failure of the statistical tests to reject the null hypothesis at low sample sizes. However, despite the presence of spurious patches of Poisson rainfall there are also many examples of patches that are likely to be genuine representations of the Poisson distribution, such as in Fig. 3. It is difficult to differentiate between these patches with the statistical tests given the small sample sizes. It is also not clear whether these genuine Poisson patches occur because the homogeneous Poisson hypothesis is applicable under

certain rainfall conditions. E.g. consistent light stratiform type rainfall. Or whether, these patches arise through randomness due to the sheer number of rainfall patches tested. This should be investigated further.

These findings highlight some limitations in how rainfall is observed with ground based instruments. The intervalometer and disdrometer used in this study had a surface area of 9.6 cm$^2$ and 14.5 cm$^2$ respectively. Consequently, the number of drops that is observed is quite low and the number of 10 s drop counts for a coherence time of 2 minutes is only 12. Practically this

means that statistical tests do not have enough power to reject the null hypothesis. Furthermore, increasing the length of the coherence time is not a suitable solution. The presence of these fine structures within rainfall would be obscured by larger sampling windows.

New sampling techniques or observation methodologies are needed to increase the effective sample size. One way of increasing the number of available observations is by increasing the effective surface area of the measuring instruments. This can





be done by using many co-located instruments. In this way the number of observations per window of time could be increased and the aggregation bin could be decreased to 5 s or 1 s thus increasing the number of drop counts available for testing at very short patch lengths. The number of observations could also be increased by increasing the sensitivity of the sensors to lower drop diameters. Another possibility would be to use adaptive sampling techniques i.e., make sure each time interval has the same number of raindrops or rainfall amount, similarly to the idea proposed by Schleiss (2017). This would allow for a better

interrogation of the Poisson hypothesis on the very fine rainfall structures present in convective storms.

Despite the issue with sample size and the fact that the Poisson hypothesis is likely not strictly true, the presence of significant amounts of homogeneous Poisson rain combined with the accuracy of derived rainfall estimates found in this study is compelling evidence for retaining the Poisson model. Furthermore, as was pointed out by Jameson and Kostinski (1998) the observed presence of any non-clustering Poissonian structures in the rainfall conflicts with a fractal description of rain and is

good argument against abandoning the Poisson framework completely for a fractal description or some other model.

## 6   Conclusions

This research leads to the following conclusions.

1. The majority of rainfall and almost all the convective type rainfall, which contributed most to total rainfall amount in this study, did not exhibit characteristics that are consistent with the Poisson hypothesis. Patches that complied with the Poisson hypothesis were characterised by low mean rainfall arrival rates during periods of sustained stratiform type rainfall.

No examples of Poisson distributed rain patches with $\rho_A > 1500 \,\mathrm{m}^{-2}.\mathrm{s}^{-1}$ were observed. Changes in the mean drop arrival rate and correlations between drop counts at scales as small as 2 minutes accounted for deviations from Poisson in 52.9 % of all rainfall patches.

2. There appear to be genuine examples of Poisson rainfall that occur during consistent light stratiform type rainfall conditions. However, small sample sizes were an issue in this study and may have resulted in the statistical tests failing to reject the null hypothesis of Poisson at low arrival rates for many rainfall patches making it hard to differentiate between genuine and spurious Poisson rainfall. Increasing the patch length is not a suitable solution to increase the number of observations. Fine structures are observed in rainfall at very small scales and sampling across such structures with different

means may actually lead to increased uncertainty in the mean. New sampling techniques or observation methodologies are needed to increase the effective sample size.

3. Total cumulative rainfall estimates derived from the disdrometer drop counts with the Marshall and Palmer (1948) parameterisation were 2 % of co-located tipping bucket measurements.




4. The intervalometers at both tipping bucket sites largely over-estimated the total rainfall amount compared with the gauges. This was most likely due to a poor calibration of the parameter $D_{\min}$. Constraining the intervalometer arrival rates by the disdrometer observed mean drop sizes results in rainfall estimates that are within 1 % of tipping bucket measurements. The accuracy of rainfall estimates is determined by the validity of the DSD parameterisation as well as the accuracy of the sensor.

5. It is possible to retrieve rainfall rates using an intervalometer. For best performances and accurate retrievals of mean drop diameters, the intervalometer needs to be used in conjunction with co-located rain gauges. The intervalometer principle shows good potential for providing ground based rainfall observations in remote areas of Africa. The main advantage of this instrument is its low cost. However, further improvements are needed to make the sensor more robust as several instruments were damaged by water during this study. Our results also show that more efforts need to be invested in determining the minimum measurable drop diameter by the intervalometer before wide scale deployment can be considered.

*Code and data availability.* http://resolver.tudelft.nl/uuid:4aad89f6-ac52-4e46-be5f-e5137f6b31c3

*Author contributions.* RH and NG contributed to the designs of the disdrometer and intervalometer. NG and MV were responsible for acquiring funding. DV and NG designed the experiment. Data was collected by DV. Analysis was performed by DV with contributions from NG, MC and MV. DV prepared the draft of the manuscript with contributions from all the co-authors.

*Competing interests.* No competing interests are present.

*Acknowledgements.* The work leading to these results has received funding from the European Community's Horizon 2020 Programme (2014-2020) under grant agreement No. 776691 (TWIGA). The opinions expressed in the document are of the authors only and no way reflect the European Commission's opinions. The European Union is not liable for any use that may be made of the information. The following are acknowledged (in no particular order) for their work in developing the disdrometer and intervalometer, Stijn de Jong, Jan Jaap Pape, Coen Degen, Ravi Bagree, Jeroen Netten, Els Veenhoven, Dirk van der Lubbe-Sanjuan, Wouter Berghuis, Rolf Hut and Nick van de Giesen. Special thanks to the hotels located on Mafia Island (Didimiza Guest House, Meremeta Lodge, Eco Shamba Kilole Lodge, Kinasi Lodge, Pole Pole Bungalows and the Mafia Island Lodge) and Chole Island (Chole Mjini Treehouse Lodge) for allowing access to their land and support in setting up the experiment.



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
