# Peer review of "Something fishy going on? Evaluating the Poisson hypothesis for rainfall estimation using intervalometers: results from an experiment in Tanzania"

_Atmospheric Measurement Techniques, 2020_

## Referee Comment (RC1) · Anonymous Referee #1 · 12 Oct 2020

This is an important contribution to the literature: a thorough and comprehensive observation study of the Poisson hypothesis for rainfall homogeneity and stationarity. It is carefully documented and diligently executed. It is, in my opinion, acceptable for publication in AMT as is.

To the extent that the study centers on testing the Poisson hypothesis via equations 18 and 19, I wish to stress the difference between the Poisson distribution and the homogeneous (stationary) Poisson process (Poisson distribution at ALL scales). The authors clearly understand that and test it on rain rates. However, testing on the basis of drop counts may also be interesting. They may want to look into the notion of

the pair-correlation function (introduced in Atmospheric Science in Kostinski, A.B. and Jameson, A.R., 2000. On the spatial distribution of cloud particles. Journal of the atmospheric sciences, 57(7), pp.901-915. See equation 5, in particular. Poisson process requires that the function (v.s. spatial or time scale) be identically zero. More importantly, it shows that Poisson distribution at a given time scale can result if there are opposing tendencies of clustering and exclusion at sub-scales.

In the context of rain, see Kostinski, Larsen, and Jameson. "The texture of rain: Exploring stochastic micro-structure at small scales." Journal of Hydrology 328.1-2 (2006): 38-45.

———————————————————

---

## Short Comment (SC1) · 16 Oct 2020

Dear reviewer thank you for taking the time to read and comment on our paper. Your comments/suggestions are appreciated. We will come back to these in more detail when the other reviewers comments have been received.

---

## Referee Comment (RC2) · Remko Uijlenhoet (Referee) · 19 Oct 2020

**Review of manuscript AMT-2020-174 "Something fishy going on? Evaluating the Poisson hypothesis for rainfall estimation using intervalometers: results from an experiment in Tanzania", submitted to AMT by Didier de Villiers et al.**

Remko Uijlenhoet, Wageningen, October 18th 2020

General remarks

- The authors present a new type of rainfall sensor (which they call "intervalometer") and show results from a field experiment using this instrument and a collocated disdrometer and tipping bucket rain gauge during monsoon rainfall in Tanzania with the aim to test the validity of the Poisson ("fish" in French) hypothesis, an implicit assumption in many models of the microstructure of rainfall, including those used for developing rainfall retrieval algorithms for remote sensors (notably weather radar). As such, the topic of this study seems to be well-suited for publication in AMT.

- The cornerstone of the rainfall estimation algorithms presented by the authors in Section 3 is the exponential drop size distribution (DSD) model (L.119, Eq.(1)), with a fixed intercept parameter $N_0$ (8000 [$mm^{-1}$ $m^{-3}$]) and a slope parameter $\Lambda$ depending on rain rate according to a power law ($\Lambda = 4.1 R^{-0.21}$ [$mm^{-1}$]), such as proposed by Marshall and Palmer (1948) (hereafter abbreviated as MP48). This model allows conversion of raindrop arrival rates observed with intervalometers first to values of $\Lambda$ (using a relationship based on the MP48 DSD model combined with a minimum detectable raindrop size $D_{min}$ of 0.8 mm). Subsequently, these $\Lambda$-values are converted to rain rates using the MP48 $\Lambda(R)$ relation (Section 3.1). The authors admit that one source of error affecting this algorithm is "model error that arises from the assumption that the DSD is adequately described by the Marshall and Palmer (1948) exponential parameterisation rather than some other parameterisation" (L.147-148). However, they indicate that "Model error will not be accounted for as the focus of this study is to test the homogeneity assumption that underlies these models rather than compare different DSD models" (L.150-152).

  Nevertheless, I expect model error to play a significant role in the reported rainfall estimates, simply because the MP48 model is -because of its fixed intercept parameter- a very special case of the general exponential DSD model, which MP48 found to adequately describe raindrop size distributions observed in Montreal, Canada (dominated by stratiform precipitation), but which may not at all be appropriate for applications in Tanzania (with rainfall likely being of a much more convective nature – see L.329-330, where you state that "this type of rainfall pattern [sustained stratiform type rainfall] is quite atypical for the rainfall record as a whole"). Previous research has indicated that in many climatic settings the value of $N_0$ cannot at all be considered constant, but shows significant variability, sometimes exhibiting a clear dependence on rain rate itself, in a similar way as $\Lambda$ depends on $R$ (e.g. Uijlenhoet, 2001, Fig.3). In general, temporal (or spatial) changes in rain rate can be caused by changes in the raindrop concentration (or flux), by changes in the (mean) raindrop size, or (and this is the most common case) by simultaneous changes in the numbers and sizes of the drops. The relative contributions of each of these depend on the type of rainfall and the climatic setting (Uijlenhoet et al., 2003, Fig.1).

  The consequence of this would be a different $\Lambda(R)$ relation than the MP48 $\Lambda(R)$ relation. If this were the case for Tanzania, this could seriously affect rain rate estimates from intervalometers if not considered in a rainfall retrieval algorithm. This does not only pertain to the algorithm based on intervalometer-observations alone (as presented in Section 3.1), but also to the extended algorithm (Section 3.2) where additional disdrometer observations are introduced to correct "for biases in the measurement of rainfall rate by the intervalometer" (L.140). That is because also the proposed bias correction is based on the MP48 $\Lambda(R)$ relation. Hence, although a number of more specific remarks concerning this issue are provided below, I would appreciate a general response to this point from the side of the authors at this point.

- Section 4.2 (Figs. 7-9) represents the core scientific contribution of this manuscript, as far as I am concerned. Although the dataset being analyzed is not very extensive, these are definitely novel, interesting and relevant results. More than a quarter of the rain collected during the measurement campaign the authors organized in Tanzania was found to represent purely Poissonian rainfall according to a total of five statistical tests, a result that I did not anticipate. For the remaining rainfall patches the authors have been able to identify which of the Poisson tests were failed. The authors find that "Poisson rain is characterised by low arrival rates" (L.342 and Fig.9) and presumably by low rain rates (although that is not explicitly shown).
- Overall, I enjoyed reading this rather lengthy but well-written paper. It introduces a new rainfall sensor, the intervalometer, which is used to estimate rainfall rates from raindrop arrival rates. As indicated above and below, I have a number of questions concerning the algorithm employed to convert drop counts to rain rates. In the second half of the paper the instrument is employed to test the Poisson hypothesis in Tanzanian rainfall. I find this the most original and convincing part of the paper. The discussion (Section 5) puts several aspects of this study into the proper perspective. In summary, I recommend moderate revisions.

Specific remarks

- L.11-12: "estimates of total rainfall amount over the entire observational period derived from disdrometer drop counts are within 2 % of co-located tipping bucket measurements". A few lines later the authors state: "Intervalometer estimates of total rainfall when corrected for minimum drop size are within 1 % of co-located tipping bucket measurements". Is it 1% or 2%?
- L.34-43: The way in which this paragraph is phrased suggests that satellites only use radars to retrieve rainfall rates. Certainly, the most advanced meteorological satellites (TRMM in the past; the GPM core satellite at this moment; CloudSat mainly for cloud observation) carry active microwave sensors (radars), but the majority only carry passive microwave sensors (radiometers) or IR sensors. That said, for all these types of sensors, a priori knowledge of the microstructure of precipitation (e.g. DSD in rainfall) is a prerequisite for developing accurate and robust rainfall retrieval algorithms. Hence, it would be appropriate to generalize the statement concerning satellite rainfall retrievals, or limit it to (ground-based) weather radars.
- L.82-84: Although I appreciate the description of the instrument provided on the indicated github page, it is a pity that the design, operation and performance of the intervalometer and of the disdrometer developed at TU Delft have not yet been published in the peer-reviewed scientific literature, as far as I can tell.
- L.101-102, "Tipping buckets were calibrated in the field [...] at a rate slower that 20 mm.h$^{-1}$ onto the instrument and recording the number of tips": The calibration of tipping bucket rain gauges if well-known to be intensity-dependent. Without taking this effect into account large rainfall rates (and hence rainfall accumulations) are typically underestimated (see Humphrey et al., 1997, and references therein). Could that phenomenon have played a role in this case?
- L.111, "Figure 2 presents an overview of the data available": Examination of Fig.2 suggests that during the entire measurement campaign of less than two months there were no time instants during which all instruments indicated in Fig.1 operated simultaneously. Was that indeed the case?
- L.134, "Uijlenhoet and Stricker (1999) showed that [...]". Note that in the derivation of these values it was tacitly assumed that the integration limits of the DSD were 0 and $\infty$, respectively. In other words, the values $\alpha = 3.25$ and $\beta = 0.762$, employed by the authors to derive their relationship between rain rate and raindrop arrival rate, are not fully consistent with the integration limits employed by the authors ($D_{min} = 0.8$ mm and $D_{max} = \infty$). Among others, the rainfall rate depends on these integration limits in much the same way as the raindrop arrival rate (Eq.(6), L.133). This will undoubtedly affect subsequent analyses presented by the authors.
- L.138-139, "The rainfall rate ($R$) can be estimated by re-arranging the Marshall and Palmer (1948) $\Lambda-R$ relation": Apart from the issue with the integration limits (see previous remark) one could question the applicability of the Marshall-Palmer DSD parameterization (derived from DSD

measurements under mostly stratiform conditions in Montreal, Canada) to raindrop arrival rate measurements from Tanzania, for rainstorms with a presumably much more convective nature (as exemplified by Fig.8).

- L.148-149, "Many alternative models for the DSD have been proposed and tested in the literature of which the most widely used are the exponential [...]": However, the Marshall-Palmer parameterization is a particular form of the general exponential DSD model. Therefore, the exponential model cannot really be termed an "alternative model" compared to the Marshall-Palmer parameterization.

- L.155-156, "the probability distribution of the drop diameters arriving at a surface per unit time [...] is a gamma distribution": Note that this statement is not true in general, but only if the DSD (in a volume) is of the exponential form ánd if the $v(D)$-relation is a power law. If the DSD (in a volume) is of the gamma form (and the $v(D)$-relation is a power law), then de pdf of the drop diameters at a surface will still be a gamma distribution, but with different parameters than Eq.(7).

- L.162, Eq.(9): This is an unnecessarily complicated form of the expression for the expected raindrop diameter at the surface. By definition, the expected value of $D$ is the first order moment of Eq.(7), i.e. the integral (between $D_{min}$ and $\infty$) of D times the pdf. Direct calculation of this integral leads to: $\mu_D = \Gamma(2+\beta,\Lambda D_{min}) / \Gamma(1+\beta,\Lambda D_{min}) \Lambda^{-1}$, which reduces to $(1+\beta) / \Lambda$ if $D_{min} = 0$. As an added value, using this expression will also simplify Eqs.(11), (12), (13) and (16) significantly.

- L.164: You probably add the subscript "exp" to $\mu_D$, $D_A$ and $\Lambda$ to contrast them with their observed ("obs") values a few sentences later (L.166-167). However, this may lead to confusion for the reader because "exp" could be interpreted as meaning both "exponential" and "expected". Therefore, please add a short statement explaining why you introduce this notation here and what the intended meaning of the subscript "exp" is.

- L.192-194: To compensate for "a small time delay between the first sensing of a raindrop with the intervalometer and the first tip registered by a tipping bucket [...] rainfall data is aggregated over an entire day". Indeed, for a steady rain rate of 1 mm/h a tipping bucket rain gauge with a resolution of 0.2 mm tips on average 5 times per hour (i.e. once every 12 min). At 12 mm/h this is once per min. So the mentioned "small time delay" can indeed be of the order of a minute or more. However, why immediately aggregate to an entire day rather than to, say, one hour? That would definitely add relevant hydrological information for potential end users.

- Figs.4 and 5, Table 1: You mainly evaluate the performance of the intervalometer as compared to the tipping bucket rain gauge in terms of rainfall accumulations over several rainfall events. However, although getting the rainfall volumes right is important, for many applications being able to capture the intra-event the dynamics is also of relevance. I wonder what scatterplots at, say, the hourly (or even shorter) time scale would look like and what the corresponding correlation coefficients would be. They could also be important performance indicators of the new intervalometer.

- Fig.5: It seems to me that it is important to stress that the corrected rain rate estimates in the left-hand panels (for the Pole Pole site) correspond to the *calibration* dataset, whereas the corrected rain rate estimates in the right-hand panels (for the MIL1 site) represent the independent *validation* dataset, even if covering only a few events.

- Fig.5: At what time scale is the correction carried out, at the event scale, the daily scale, or for the entire campaign as a whole? I could not grasp this clearly from the text, but perhaps I overlooked something.

- Fig.6, caption: Are these indeed "Estimated mean drop sizes" or would the term "effective mean drop sizes" be better? After all, they correspond to the *effective* mean drop sizes that bring the intervalometer-derived rain rates in accordance with those from the tipping bucket rain gauge, if I have understood correctly.

- Fig.6: At least as informative from the perspective of rain rate estimation are the *volume-weighted mean diameter* and the *median-volume diameter* (see Uijlenhoet and Stricker, 1999, Table 4). They are more indicative (than the mean diameter shown in Fig.6) of the *effective* drop size contributing most to the rain rate. An advantage with respect to the mean diameter (i.e. the

expected value corresponding to the DSD itself) is that these characteristic diameters are less sensitive to the small droplet end of the size spectrum, hence much less prone to the issue of instrumental truncation at some lower diameter limit ($D_{min}$).

- L.312-313, "Overall, only 28.3 % of all patches can reasonably be assumed to be Poisson distributed": I would not call this "only 28.3 %". Apparently, more than a quarter of the rainfall during the experiment "are patches of stationary rainfall that exhibit no correlation between drop counts within a 95 % confidence interval, match a Poisson distribution very well and have a mean dispersion of approximately 1" (L.313-314). If you would have asked me in advance, I would not have guessed that such a significant fraction of all rain collected during your campaign would pass *all* of the tests you outlined in Section 3.4.
- L.341, "random sampling effects": For purely Poissonian rainfall closed-form expressions for the strength of such sampling fluctuations can be derived (e.g. Uijlenhoet et al., 2006). There is no need to incorporate this reference in your manuscript. It is just provided for your information.
- Fig.9: These are interesting results. I wonder what these plots would have looked like if you would have used rain rate [mm/h] on the x-axis rather than the raindrop arrival rate. Regarding the bottom panel, it seems it would be good to refer back to Fig.6 somewhere in the last paragraph of Section 4.2 (L.357-366), because both are plots of mean raindrop diameter versus drop arrival rate.
- L.402-403: The authors conclude that "Marshall and Palmer's (1948) parameterisation of the DSD is a good approximation of the observed DSD over the period of the experiment". However, they have not explicitly tested this conclusion using the available disdrometer observations. Therefore, it seems that the evidence for this conclusion could have been stronger than it currently is (based only on the "derivation of reasonably accurate rainfall measurements", L.401-402).

Editorial remarks

- L.24: Insert comma before "but progress ...".
- L.25: The reference "(TAH, 2017)" is not listed in the reference list.
- L.28, 33, 34, 62, 63, etc.: Insert hyphen between "ground" and "based".
- L.41: Replace closing quotation mark before "truthing" with opening quotation mark.
- L.53: Replace semicolon by colon.
- L.54: Replace closing quotation mark before "streakiness" with opening quotation mark.
- L.56: Replace "like" by "e.g." and put it after the opening parenthesis of the citation.
- L.57: Insert hyphen between "scale" and "dependent".
- L.58: Remove comma after "Others".
- L.64: Insert hyphen between "two" and "month".
- L.64: Insert "The" before "Marshall and Palmer".
- L.69: Insert comma before "respectively".
- L.74: Remove comma before "in the US".
- L.85, 86: "at e.g." --> "in e.g." (2x); to put parentheses only around publication years, use \citet instead of \citep for references in LaTeX.
- L.88: Insert "typically" before "0.2 mm".
- L.120: This should not be listed as a separate equation (Eq.(2)), but rather as line in the main body of the text.
- L.130: Correct "0.5 mm ≥ D ≤ 5.0 mm" to "0.5 mm ≤ D ≤ 5.0 mm".
- L.130: Correct "The mean rainfall arrival rate" to "The mean raindrop arrival rate".
- L.131: Add "mm" after "0.8".
- L.134-135: Insert hyphen between "self" and "consistency".
- L.146: Insert comma after "sensor".
- L.149: Insert comma after "literature".
- L.154: "constrain" --> "constraint".
- L.157: Replace the full stop at the end of the sentence with a colon.

- L.159: This should not be listed as a separate equation (Eq.(8)), but rather as a specification of the constraints on the parameters of Eq.(7).
- L.166: Insert comma before "these can be". For clarity, the mathematical notation "$f(\rho_{A,obs})$" may even be omitted completely, because it is already described in words ("Now, if the observed mean drop sizes [...] are some function of [...]").
- L.170: Replace closing quotation mark before "corrected" with opening quotation mark.
- L.171: Replace the full stop at the end of the sentence with a colon.
- L.172-175, Eqs.(11-13): Here, the exponent -1/0.21 is employed, whereas on L.139 the exponent -4.762 is used. Please use the same notation.
- L.179: Insert space after "e.g.".
- L.181, 191: Replace closing quotation mark before "corrected" with opening quotation mark.
- L.186: Replace the full stop at the end of the sentence with a colon.
- L.190, "Expected mean drop size": Either use "expected" or "mean".
- L.198: Replace the reference to Uijlenhoet and Stricker (1999) by a reference to Uijlenhoet et al. (1999).
- L.205: "(eg. (Feller, 2010))" --> "(e.g. Feller, 2010)".
- L.205: Replace the full stop at the end of the sentence with a colon.
- L.207: "Where" --> "where".
- L.212: Replace the full stop at the end of the sentence with a colon.
- L.217: "homogeneous" --> "homogeneity".
- L.225: Insert "interarrival" before "time".
- L.238: Insert a comma before "whereas".
- L.239-240: Put parentheses only around publication years, i.e. use \citet instead of \citep for references in LaTeX.
- L.249: L.198: Replace the reference to Uijlenhoet and Stricker (1999) by a reference to Uijlenhoet et al. (1999).
- L.272: "Rates" --> "rates".
- L.274, caption Fig.4: Replace closing quotation mark before "online" with opening quotation mark (similar for "Exp" and "Corr").
- Caption Fig.3: "an example" --> "a sample".
- Fig.9: Y-axis labels of top and middle panels should also read "Mean Drop Diameter [mm]", because they represent mean diameters for each 10 s drop count.
- L.377: "over estimated" --> "overestimated".
- L.378: Insert comma before "which".
- L.378: "over-estimated" --> "overestimated".
- L.393: Insert comma before "which".
- L.395: Insert hyphen between "disdrometer" and "observed".
- L.426: "aim" --> "aims".
- L.443: Insert comma before "which".
- L.447: Insert comma before "such".
- L.448: Insert comma after "However".
- L.452: Insert comma before "making".
- L.460, "E.g.", "Or whether": This should be one sentence, replacing the full stop after "rainfall conditions" with a comma.
- L.462: "ground based" --> "ground-based".
- L.463: Insert comma before "respectively".
- L.471: Insert comma before "thus".
- L.478: Insert comma after "(1998)".
- L.482: Replace full stop at end of sentence with colon.
- L.492: Insert comma after "patches".
- L.509: "ground based" --> "ground-based".
- L.512: "wide scale" --> "wide-scale".
- L.528: In the reference list, the titles of books should be written with all nouns capitalized; the titles of papers only with the first word of the title capitalized (except for proper nouns).

- L.529-532: The first three references in the reference list have no author and do not appear in alphabetical order. Please correct.
- L.558-560: The correct journal name, issue number, page numbers and doi for this reference are: *Hydrol. Earth Syst. Sci.*, **23**, 4737–4761, doi:10.5194/hess-23-4737-2019.
- L.610: Write "Poisson" with a capital "P".

References (if not yet listed in the manuscript)

- Humphrey, M.D., J.D. Istok, J.Y. Lee, J.A. Hevesi, and A.L. Flint, 1997: A new method for automated dynamic calibration of tipping-bucket rain gauges. *J. Atmos. Oceanic Technol.*, **14**, 1513–1519, doi:10.1175/1520-0426(1997)014<1513:ANMFAD>2.0.CO;2.
- Uijlenhoet, R., 2001: Raindrop size distributions and radar reflectivity–rain rate relationships for radar hydrology. *Hydrol. Earth Syst. Sci.*, **5**, 615–627, doi:10.5194/hess-5-615-2001.
- Uijlenhoet, R., J.M. Porrà, D. Sempere-Torres, and J.-D. Creutin, 2006: Analytical solutions to sampling effects in drop size distribution measurements during stationary rainfall: Estimation of bulk rainfall variables. *J. Hydrol.*, **328**, 65–82, doi:10.1016/j.jhydrol.2005.11.043.
- Uijlenhoet, R., J.A. Smith and M. Steiner, 2003: The microphysical structure of extreme precipitation as inferred from ground-based raindrop spectra. *J. Atmos. Sci.*, **60**, 1220–1238, doi:10.1175/1520-0469(2003)60<1220:TMSOEP>2.0.CO;2.

---

## Author Comment (AC1) · 11 Mar 2021

Response to Review Comment 1:

The authors would like to thank the reviewer for their review of our paper. The comments are very well received. Each of the reviewer's comments will be addressed one by one in the text below. The reviewer's comment is listed first followed by the authors' response to the comment in red and finally the proposed changes to the paper (if any).

1. *This is an important contribution to the literature: a thorough and comprehensive observation study of the Poisson hypothesis for rainfall homogeneity and stationarity. It is carefully documented and diligently executed. It is, in my opinion, acceptable for publication in AMT as is.*
2. Thank you
3. No changes

1. *To the extent that the study centers on testing the Poisson hypothesis via equations 18 and 19, I wish to stress the difference between the Poisson distribution and the homogeneous (stationary) Poisson process (Poisson distribution at ALL scales). The authors clearly understand that and test it on rain rates. However, testing on the basis of drop counts may also be interesting. They may want to look into the notion of the pair-correlation function (introduced in Atmospheric Science in Kostinski, A.B. and Jameson, A.R., 2000. On the spatial distribution of cloud particles. Journal of the atmospheric sciences, 57(7), pp.901-915. See equation 5, in particular. Poisson process requires that the function (v.s. spatial or time scale) be identically zero. More importantly, it shows that Poisson distribution at a given time scale can result if there are opposing tendencies of clustering and exclusion at sub-scales.*
2. Thank you for the suggested reading. The authors would like to clarify that the assumption of homogeneity is tested on the 10s drop counts and not on rainfall rates.
3. No changes

1. *In the context of rain, see Kostinski, Larsen, and Jameson. "The texture of rain: Exploring stochastic micro-structure at small scales." Journal of Hydrology 328.1-2 (2006): 38-45.*
2. Thank you for the suggestion.
3. No changes

---

## Author Comment (AC2) · 11 Mar 2021

Response to Review Comment 2:

The authors would like to thank Remko Uijlenhoet for his in-depth review of our paper. The comments are very well received, and we believe that in addressing the issues he has raised, the overall quality of the paper has been improved. Each of Remko's comments will be addressed one by one in the text below. The reviewer's comment is listed first followed by the authors' response to the comment in red and finally the proposed changes to the paper (if any).

***General remarks***

1. *The authors present a new type of rainfall sensor (which they call "intervalometer") and show results from a field experiment using this instrument and a collocated disdrometer and tipping bucket rain gauge during monsoon rainfall in Tanzania with the aim to test the validity of the Poisson ("fish" in French) hypothesis, an implicit assumption in many models of the microstructure of rainfall, including those used for developing rainfall retrieval algorithms for remote sensors (notably weather radar). As such, the topic of this study seems to be well-suited for publication in AMT.*
2. Thank you.
3. No changes

1. *The cornerstone of the rainfall estimation algorithms presented by the authors in Section 3 is the exponential drop size distribution (DSD) model (L.119, Eq.(1)), with a fixed intercept parameter $N_0$ (8000 [$mm^{-1} m^{-3}$]) and a slope parameter $\Lambda$ depending on rain rate according to a power law ($\Lambda = 4.1 R^{-0.21}$ [$mm^{-1}$]), such as proposed by Marshall and Palmer (1948) (hereafter abbreviated as MP48). This model allows conversion of raindrop arrival rates observed with intervalometers first to values of $\Lambda$ (using a relationship based on the MP48 DSD model combined with a minimum detectable raindrop size $D_{min}$ of 0.8 mm). Subsequently, these $\Lambda$-values are converted to rain rates using the MP48 $\Lambda(R)$ relation (Section 3.1). The authors admit that one source of error affecting this algorithm is "model error that arises from the assumption that the DSD is adequately described by the Marshall and Palmer (1948) exponential parameterisation rather than some other parameterisation" (L.147-148). However, they indicate that "Model error will not be accounted for as the focus of this study is to test the homogeneity assumption that underlies these models rather than compare different DSD models" (L.150-152).*
*Nevertheless, I expect model error to play a significant role in the reported rainfall estimates, simply because the MP48 model is -because of its fixed intercept parameter- a very special case of the general exponential DSD model, which MP48 found to adequately describe raindrop size distributions observed in Montreal, Canada (dominated by stratiform precipitation), but which may not at all be*

*appropriate for applications in Tanzania (with rainfall likely being of a much more convective nature – see L.329-330, where you state that "this type of rainfall pattern [sustained stratiform type rainfall] is quite atypical for the rainfall record as a whole"). Previous research has indicated that in many climatic settings the value of N0 cannot at all be considered constant, but shows significant variability, sometimes exhibiting a clear dependence on rain rate itself, in a similar way as Λ depends on R (e.g. Uijlenhoet, 2001, Fig.3). In general, temporal (or spatial) changes in rain rate can be caused by changes in the raindrop concentration (or flux), by changes in the (mean) raindrop size, or (and this is the most common case) by simultaneous changes in the numbers and sizes of the drops. The relative contributions of each of these depend on the type of rainfall and the climatic setting (Uijlenhoet et al., 2003, Fig.1). The consequence of this would be a different Λ(R) relation than the MP48 Λ(R) relation. If this were the case for Tanzania, this could seriously affect rain rate estimates from intervalometers if not considered in a rainfall retrieval algorithm. This does not only pertain to the algorithm based on intervalometer-observations alone (as presented in Section 3.1), but also to the extended algorithm (Section 3.2) where additional disdrometer observations are introduced to correct "for biases in the measurement of rainfall rate by the intervalometer" (L.140). That is because also the proposed bias correction is based on the MP48 Λ(R) relation. Hence, although a number of more specific remarks concerning this issue are provided below, I would appreciate a general response to this point from the side of the authors at this point.*

2. The reviewer raises a valid concern that the parameterisation of the drop size distribution (DSD) with a fixed intercept parameter N0 (8000 [$mm^{-1}$ $m^{-3}$]) and a slope parameter Λ depending on rain rate according to a power law (Λ = 4.1 $R^{-0.21}$ [$mm^{-1}$]), such as proposed by Marshall and Palmer (1948) (hereafter abbreviated as MP48) derived from stratiform rainfall in Montreal, Canada may not be applicable in Tanzanian rainfall, which is of a largely convective nature. This was investigated by comparing the observed drop size distribution to the MP48 parameterisation. During this process some discrepancies in the dataset of the disdrometer were discovered that had resulted in incorrect estimates of rainfall amount for the MP48 parameterisation. These have been investigated very thoroughly and it was determined that the minimum detectable drop size of the instrument is 1 mm which is approximately double what was first assumed based on prior work.

Figure 1 presents the observed raindrop size distribution per unit volume of air, $N_V(D)$ [$mm^{-1}m^{-3}$] as well as the MP48 parameterisation, the line of best fit and the line of best fit adjusted such that it is self-consistent using the framework presented by Uijlenhoet et al. (1999). Note that the self-consistency framework presented by Uijlenhoet et al. (1999) assumes no truncation of the drop size distribution. Since the instruments used in this research have a minimum detectable drop size it is

necessary to account for the effect of truncation in the self-consistency relationship. Equation 61 in Uijlenhoet et al. (1999) is adjusted such that:

$$R = 6 \times 10^{-4} \pi \alpha N_0 \frac{\Gamma(4+\beta, \Lambda Dmin)}{\Lambda^{4+\beta}} \quad [\text{mm.hr}^{-1}] \qquad \text{Eq 1}$$

Equation 1 can be used to determine the self-consistent value of either $N_0$ or $\Lambda$ as long as one of these parameters ($N_0$ or $\Lambda$) is known or its power law relation to R is known. This is done by entering the known ($N_0$ or $\Lambda$) and calculating the unknown parameter ($N_0$ or $\Lambda$) for different rainfall rates such that the left and right hand sides of the equation are equal. I.e., R is equal to R. The result is either a constant or a power law relation with R.

From the observed drop size distribution presented in figure 1 the value of $N_0$ is determined to be a constant value of 4342 [$\text{mm}^{-1}$ $\text{m}^{-3}$] through a linear fit of D vs ln($N_v$(D)). Using the self-consistency requirement presented in Eq. 1 determines that $\Lambda$ = 3.56 $R^{-0.204}$, the self-consistent parameterisation is plotted in Figure 1 as well. Note that the MP48 self-consistent $\Lambda$-R relation for a left truncated DSD at 1mm is $\Lambda$ = 4.06 $R^{-0.203}$. Figure 1 clearly shows that the MP48 parameterisation is a poor approximation of the observed drop size distribution and therefore rainfall rates derived from MP48 are not physically based on the observed rain drops. Therefore, it is necessary to derive a new model for the DSD to fit the data in Tanzania.

[Figure]

Figure 1. The observed drop sized distribution as well as the best fit parameterisation, the MP48 parameterisation and the self-consistent best fit parameterisation.

An obvious candidate is the self-consistent fit to the observed drop size distribution of the rainfall record as a whole presented in Figure 1. However, closer interrogation of the different rainfall events within the observation period of the disdrometers

shows that $N_0$ is not constant during the observational period and varies greatly between rainfall events. This finding is in line with the $N_0$ jumps that Waldvogel (1974) observed even within a single rainfall event. During the observational period $N_0$ varies from approximately $1000 - 12000$ [mm$^{-1}$ m$^{-3}$] over different rainfall events. The natural logarithm $ln(N_V(D))$ of the DSD is plotted against Drop Diameter [mm] in Figure 2 for each separate rainfall event with a total rain height of more than 5mm (i.e. for different rainfall rates) as well as the linear line of best fit to the data. Data have a R$^2$ value greater than 0.98.

[Figure]

Figure 2. The natural logarithm of $N_V(D)$) is plotted against diameter for rainfall events with a total rain height of more than 5 mm. The line of best fit is plotted for each rain event as well.

The slope of the line of best fit is equal to $-\Lambda$ and the intercept is $N_0$. Figure 2 clearly shows that the values of $\Lambda$ and $N_0$ differ as a function of the rainfall rate. The values of $\Lambda$ can be plotted against rainfall rate and fitted to a power law function of the rainfall rate and result in the relation, $\Lambda = 4.13R^{-0.32}$. Solving for the self-consistent $N_0$ using Equation 1 gives the following power law relation, $N_0 = 5310R^{-0.366}$. This relation can be used to derive rainfall rates and the results are plotted in Figure 3. Note that the derived power laws are quite similar to other parameterisations presented in the literature such as by Joss et al (1968) for rainfall that is a mixture of thunderstorms and widespread rain.

[Figure]

Figure 3. The total rainfall amount [mm] observed by the co-located tipping bucket, intervalometer and disdrometer at the main site (Pole Pole) for the longest 'online' period of the three instruments. Also plotted are the rainfall arrival rates measured by the disdrometer and intervalometer and the rainfall rates derived from different parameterisations of the DSD.

In Figure 3 the subscript MP48 refers to the Marshall and Palmer parameterisation; N0 refers to the experimentally derived and self-consistent N0 = 4342, Λ = 3.56 R$^{-0.204}$ parameterisation; Power Law refers to the experimentally determined parameterisation with a power law for N0, $N_0 = 5310R^{-0.366}, \Lambda = 4.13R^{-0.32}$ and Power Law Corr refers to the corrected Power Law to align the intervalometer with the disdrometer since the Power Law was derived using the disdrometer observed drop sizes

The proposed power law model for N0 leads to very accurate results for the total rainfall amount for the disdrometer, within 4% of the collocated tipping bucket measurements. After correction of the intervalometer rainfall estimates using the correction proposed in the paper the intervalometer power law also results in very accurate rainfall estimates, within 4% of the collocated tipping bucket measurements. The other parameterisations do not result in good rainfall estimates and in particular the MP48 parameterisation results in a large underestimate of the total rainfall amount.

The authors accept that the use of the MP48 parameterisation is not justified given that it does not fit the observed drop sizes as shown in Figure 1. A new Marshall and

Palmer type parameterisation has been derived based on the drop size observations with $\Lambda = 4.13R^{-0.32}$ and $N_0 = 5310R^{-0.366}$. This parameterisation results in good estimates of the total rainfall amount for the disdrometer. The derived power law parameterisation is quite crude due to the small dataset of disdrometer measurements and the exponents of the relation are highly sensitive to changes in assumptions (such as independent storm length or storm rain height). The disdrometer was only recording data for a total of 2 weeks. Therefore, further work would need to be done with a much larger dataset to properly calibrate the parameters of the model or derive a new model dependent on the local climatological conditions. Using limited data we have been able to derive reasonable estimates of the total rainfall amount using an intervalometer thus showing proof of concept for the instrument.

It is clear that different parameterisations of the DSD can be derived and should be derived depending on the local climatological conditions. However, it should also be pointed out that many studies have already done this, and many different models have already been proposed (Joss et al., 1968; Uijlenhoet et al., 1999; Bennet et al., 1983; Sekhon et al., 1973; Ulbrich, 1983)**.** Presenting a new Marshall and Palmer type DSD model or Gamma DSD model adds nothing new to the literature. In our opinion it is far more interesting to investigate the underlying assumption of homogeneity that is behind these kinds of models as has been presented in the second half of the paper.

In summary, the reviewer's point is accepted and we concede that the use of the MP48 parameterisation is not valid based on the actual observations of the drop size distribution. The authors propose to re-work and shorten the section of the paper that presents the rainfall rates to incorporate the new DSD model and the associated results to illustrate the potential of deriving accurate rainfall rates with an intervalometer but stressing that calibration of the DSD model is still a necessity. This will place the focus of the paper on the Poisson analysis.

3. All the proposed changes to the paper will be presented in the revised manuscript.

*Specific remarks*

1. *Section 4.2 (Figs. 7-9) represents the core scientific contribution of this manuscript, as far as I am concerned. Although the dataset being analyzed is not very extensive, these are definitely novel, interesting and relevant results. More than a quarter of the rain collected during the measurement campaign the authors organized in Tanzania was found to represent purely Poissonian rainfall according to a total of five statistical tests, a result that I did not anticipate. For the remaining rainfall patches the authors have been able to identify which of the Poisson tests were failed.*

*The authors find that "Poisson rain is characterised by low arrival rates" (L.342 and Fig.9) and presumably by low rain rates (although that is not explicitly shown). Overall, I enjoyed reading this rather lengthy but well-written paper. It introduces a new rainfall sensor, the intervalometer, which is used to estimate rainfall rates from raindrop arrival rates. As indicated above and below, I have a number of questions concerning the algorithm employed to convert drop counts to rain rates. In the second half of the paper the instrument is employed to test the Poisson hypothesis in Tanzanian rainfall. I find this the most original and convincing part of the paper. The discussion (Section 5) puts several aspects of this study into the proper perspective. In summary, I recommend moderate revisions.*

2. Thank you for your insightful comments. The authors agree that the Poisson findings are the core findings of this paper, as such we propose to shorten the section on rainfall rates to place the emphasis on the Poisson results. Please see above for our general response addressing the algorithm employed in deriving rainfall rates.

3. See the section above for the proposed changes to the paper

1. *L.11-12: "estimates of total rainfall amount over the entire observational period derived from disdrometer drop counts are within 2 % of co-located tipping bucket measurements". A few lines later the authors state: "Intervalometer estimates of total rainfall when corrected for minimum drop size are within 1 % of co-located tipping bucket measurements". Is it 1% or 2%?*

2. Thank you for pointing out the discrepancy.

3. These numbers have changed and now it is 4%. This will be amended in the text

1. *L.34-43: The way in which this paragraph is phrased suggests that satellites only use radars to retrieve rainfall rates. Certainly, the most advanced meteorological satellites (TRMM in the past; the GPM core satellite at this moment; CloudSat mainly for cloud observation) carry active microwave sensors (radars), but the majority only carry passive microwave sensors (radiometers) or IR sensors. That said, for all these types of sensors, a priori knowledge of the microstructure of precipitation (e.g. DSD in rainfall) is a prerequisite for developing accurate and robust rainfall retrieval algorithms. Hence, it would be appropriate to generalize the statement concerning satellite rainfall retrievals, or limit it to (ground-based) weather radars.*
*L.82-84: Although I appreciate the description of the instrument provided on the indicated github page, it is a pity that the design, operation and performance of the intervalometer and of the disdrometer developed at TU Delft have not yet been published in the peer-reviewed scientific literature, as far as I can tell.*

*2.* Although the design for the intervalometer has not been published in the peer-reviewed scientific literature a similar instrument in terms of acoustic sensor, has been presented in the PhD thesis of Hut 2013 [https://doi.org/10.4233/uuid:48d09fb4-4aba-4161-852d-adf0be352227]. Our intent is to make this publication the first publication in the scientific literature concerning an intervalometer.

*3.* Hut's PhD thesis will be referenced in the text.

1. *L.101-102, "Tipping buckets were calibrated in the field [...] at a rate slower that 20 mm.h$^{-1}$ onto the instrument and recording the number of tips": The calibration of tipping bucket rain gauges if well-known to be intensity-dependent. Without taking this effect into account large rainfall rates (and hence rainfall accumulations) are typically underestimated (see Humphrey et al., 1997, and references therein). Could that phenomenon have played a role in this case?*

2. Thank you.

3. The following sentences will be added. "High rainfall rates and associated rainfall accumulations are typically underestimated (see Humphrey et al., 1997, and references therein). This may have contributed to the observed results."

1. *L.111, "Figure 2 presents an overview of the data available": Examination of Fig.2 suggests that during the entire measurement campaign of less than two months there were no time instants during which all instruments indicated in Fig.1 operated simultaneously. Was that indeed the case?*

2. That is technically correct. However, MIL3 was only installed 2 months into the observation period as it was originally kept as a spare instrument in case any of the others went offline. Therefore, for approximately two weeks at the beginning of the observational period all installed instruments were online together.

3. No changes

1. *L.134, "Uijlenhoet and Stricker (1999) showed that [...]". Note that in the derivation of these values it was tacitly assumed that the integration limits of the DSD were 0 and ∞, respectively. In other words, the values α = 3.25 and ϐ = 0.762, employed by the authors to derive their relationship between rain rate and raindrop arrival rate, are not fully consistent with the integration limits employed by the authors (Dmin = 0.8 mm and Dmax = ∞). Among others, the rainfall rate depends on these integration limits in much the same way as the raindrop arrival rate (Eq.(6), L.133). This will undoubtedly affect subsequent analyses presented by the authors.*

2. The self-consistency framework presented by Uijlenhoet et al. (1999) has been adjusted to account for the truncation of the DSD and is presented in Eq 1 of the general response. This equation is used to re-calculate the self-consistent $\Lambda - R$ relationship using the Atlas and Ulbrich (1977) values for α, β values. For a Dmin = 1 mm the MP48 parameterisation becomes $\Lambda = 4.06\ R^{-0.203}$.

3. *This is discussed in the text.*
* * *
1. *L.138-139, "The rainfall rate (R) can be estimated by re-arranging the Marshall and Palmer (1948) $\Lambda-R$ relation": Apart from the issue with the integration limits (see previous remark) one could question the applicability of the Marshall-Palmer DSD parameterization (derived from DSD measurements under mostly stratiform conditions in Montreal, Canada) to raindrop arrival rate measurements from Tanzania, for rainstorms with a presumably much more convective nature (as exemplified by Fig.8).*

2. Please see above for our general response addressing the algorithm employed in deriving rainfall rates.

3. *No changes*
* * *
1. *L.148-149, "Many alternative models for the DSD have been proposed and tested in the literature of which the most widely used are the exponential [...]": However, the Marshall- Palmer parameterization is a particular form of the general exponential DSD model. Therefore, the exponential model cannot really be termed an "alternative model" compared to the Marshall- Palmer parameterization.*

2. The ambiguity in the wording is noted.

3. *Propose to change to: "Many models for the DSD have been proposed and tested in the literature of which the most widely used are the exponential (of which the Marshall and Palmer model is a special case)…"*
* * *
1. *L.155-156, "the probability distribution of the drop diameters arriving at a surface per unit time [...] is a gamma distribution": Note that this statement is not true in general, but only if the DSD (in a volume) is of the exponential form ánd if the v(D)-relation is a power law. If the DSD (in a volume) is of the gamma form (and the v(D)-relation is a power law), then de pdf of the drop diameters at a surface will still be a gamma distribution, but with different parameters than Eq.(7).*

2. Thank you for pointing out this detail.

3. We will include this in the text: "the probability distribution of the drop diameters arriving at a surface per unit time […] is a gamma distribution if the DSD (in a volume) is of the exponential form (e.g. Marshall and Palmer) and if the v(D) relation is a power law."

1. *L.162, Eq.(9): This is an unnecessarily complicated form of the expression for the expected raindrop diameter at the surface. By definition, the expected value of D is the first order moment of Eq.(7), i.e. the integral (between Dmin and ∞) of D times the pdf. Direct calculation of this integral leads to: $\mu_D = \Gamma(2+\beta, \Lambda D_{min}) / \Gamma(1+\beta, \Lambda D_{min}) \Lambda^{-1}$, which reduces to $(1+\beta) / \Lambda$ if Dmin = 0. As an added value, using this expression will also simplify Eqs.(11), (12), (13) and (16) significantly.*
2. *Thank you for pointing out this short-cut, which is indeed much simpler than the form presented by (Johnson et al., 2011). It will be included in the paper.*
3. *Equation 9 will be replaced by the shorter version.*

1. *L.164: You probably add the subscript "exp" to µD, DA and Λ to contrast them with their observed ("obs") values a few sentences later (L.166-167). However, this may lead to confusion for the reader because "exp" could be interpreted as meaning both "exponential" and "expected". Therefore, please add a short statement explaining why you introduce this notation here and what the intended meaning of the subscript "exp" is.*
2. The ambiguity of exponential vs expected is noted and will be clarified in the revised manuscript.
3. We propose to add the following to the text: "The subscript exp in equations 9 through 17 represents expected in contrast to the subscript obs which represents observed and should not be confused with exponential."

1. *L.192-194: To compensate for "a small time delay between the first sensing of a raindrop with the intervalometer and the first tip registered by a tipping bucket [...] rainfall data is aggregated over an entire day". Indeed, for a steady rain rate of 1 mm/h a tipping bucket rain gauge with a resolution of 0.2 mm tips on average 5 times per hour (i.e. once every 12 min). At 12 mm/h this is once per min. So the mentioned "small time delay" can indeed be of the order of a minute or more. However, why immediately aggregate to an entire day rather than to, say, one hour? That would definitely add relevant hydrological information for potential end users.*
2. This section is removed from the manuscript.
3. This section is removed from the manuscript.

1. *Figs.4 and 5, Table 1: You mainly evaluate the performance of the intervalometer as compared to the tipping bucket rain gauge in terms of rainfall accumulations over several rainfall events. However, although getting the rainfall volumes right is important, for many applications being able to capture the intra-event the dynamics is also of relevance. I wonder what scatterplots at, say, the hourly (or even shorter)*

*time scale would look like and what the corresponding correlation coefficients would be. They could also be important performance indicators of the new intervalometer.*

2. Thank you, this would indeed be interesting to investigate. Based on the findings of this paper we hope to improve the rainfall retrieval of the intervalometer.

3. No changes.

1. *Fig.5: It seems to me that it is important to stress that the corrected rain rate estimates in the left-hand panels (for the Pole Pole site) correspond to the calibration dataset, whereas the corrected rain rate estimates in the right-hand panels (for the MIL1 site) represent the independent validation dataset, even if covering only a few events.*

2. This figure is no longer presented.

3. Figure 5 is removed from the manuscript.

1. *Fig.5: At what time scale is the correction carried out, at the event scale, the daily scale, or for the entire campaign as a whole? I could not grasp this clearly from the text, but perhaps I overlooked something.*

2. The correction is applied to each 10s drop count using mean values of the observed drop sizes that were derived from the entire observational period of the disdrometer.

3. No changes

1. *Fig.6, caption: Are these indeed "Estimated mean drop sizes" or would the term "effective mean drop sizes" be better? After all, they correspond to the effective mean drop sizes that bring the intervalometer-derived rain rates in accordance with those from the tipping bucket rain gauge, if I have understood correctly.*

2. Indeed, effective is probably better terminology.

3. Estimated is replaced by effective.

1. *Fig.6: At least as informative from the perspective of rain rate estimation are the volume- weighted mean diameter and the median-volume diameter (see Uijlenhoet and Stricker, 1999, Table 4). They are more indicative (than the mean diameter shown in Fig.6) of the effective drop size contributing most to the rain rate. An advantage with respect to the mean diameter (i.e. the expected value corresponding to the DSD itself) is that these characteristic diameters are less sensitive to the small droplet end of the size spectrum, hence much less prone to the issue of instrumental truncation at some lower diameter limit (Dmin).*

2. Figure 6 is no longer part of the manuscript.

3. Figure 6 is removed from this manuscript as well as any associated discussion.

1. L.312-313, "Overall, only 28.3 % of all patches can reasonably be assumed to be Poisson distributed": I would not call this "only 28.3 %". Apparently, more than a quarter of the rainfall during the experiment "are patches of stationary rainfall that exhibit no correlation between drop counts within a 95 % confidence interval, match a Poisson distribution very well and have a mean dispersion of approximately 1" (L.313-314). If you would have asked me in advance, I would not have guessed that such a significant fraction of all rain collected during your campaign would pass all of the tests you outlined in Section 3.4.

2. We also found this result to be surprising.

3. The word "only" will be removed

1. L.341, "random sampling effects": For purely Poissonian rainfall closed-form expressions for the strength of such sampling fluctuations can be derived (e.g. Uijlenhoet et al., 2006). There is no need to incorporate this reference in your manuscript. It is just provided for your information.

2. Thank you

3. No changes

1. Fig.9: These are interesting results. I wonder what these plots would have looked like if you would have used rain rate [mm/h] on the x-axis rather than the raindrop arrival rate. Regarding the bottom panel, it seems it would be good to refer back to Fig.6 somewhere in the last paragraph of Section 4.2 (L.357-366), because both are plots of mean raindrop diameter versus drop arrival rate.

2. Figure 6 has been removed from the manuscript.

3. No changes

1. L.402-403: The authors conclude that "Marshall and Palmer's (1948) parameterisation of the DSD is a good approximation of the observed DSD over the period of the experiment". However, they have not explicitly tested this conclusion using the available disdrometer observations. Therefore, it seems that the evidence for this conclusion could have been stronger than it currently is (based only on the "derivation of reasonably accurate rainfall measurements", L.401- 402).

2. The conclusion was incorrect. Please see above for our general response addressing the algorithm employed in deriving rainfall rates.

3. This incorrect conclusion will be removed.

**Editorial remarks**

The suggested editorial remarks will be incorporated into the revised manuscript if they represent an obvious error. The authors retain the right to disagree on grammar.

- *L.24: Insert comma before "but progress ...".*
- *L.25: The reference "(TAH, 2017)" is not listed in the reference list.*
- *L.28, 33, 34, 62, 63, etc.: Insert hyphen between "ground" and "based".*
- *L.41: Replace closing quotation mark before "truthing" with opening quotation mark.*
- *L.53: Replace semicolon by colon.*
- *L.54: Replace closing quotation mark before "streakiness" with opening quotation mark.*
- *L.56: Replace "like" by "e.g." and put it after the opening parenthesis of the citation.*
- *L.57: Insert hyphen between "scale" and "dependent".*
- *L.58: Remove comma after "Others".*
- *L.64: Insert hyphen between "two" and "month".*
- *L.64: Insert "The" before "Marshall and Palmer".*
- *L.69: Insert comma before "respectively".*
- *L.74: Remove comma before "in the US".*
- *L.85, 86: "at e.g." --> "in e.g." (2x); to put parentheses only around publication years, use \citet instead of \citep for references in LaTeX.*
- *L.88: Insert "typically" before "0.2 mm".*
- *L.120: This should not be listed as a separate equation (Eq.(2)), but rather as line in the main body of the text.*
- *L.130: Correct "0.5 mm ≥ D ≤ 5.0 mm" to "0.5 mm ≤ D ≤ 5.0 mm".*
- *L.130: Correct "The mean rainfall arrival rate" to "The mean raindrop arrival rate".*
- *L.131: Add "mm" after "0.8".*
- *L.134-135: Insert hyphen between "self" and "consistency".*
- *L.146: Insert comma after "sensor".*
- *L.149: Insert comma after "literature".*
- *L.154: "constrain" --> "constraint".*
- *L.157: Replace the full stop at the end of the sentence with a colon.*

- *L.159: This should not be listed as a separate equation (Eq.(8)), but rather as a specification of the constraints on the parameters of Eq.(7).*
- *L.166: Insert comma before "these can be". For clarity, the mathematical notation "f(ρA,obs)" may even be omitted completely, because it is already described in words ("Now, if the observed mean drop sizes [...] are some function of [...]").*
- *L.170: Replace closing quotation mark before "corrected" with opening quotation mark.*
- *L.171: Replace the full stop at the end of the sentence with a colon.*

- *L.172-175, Eqs.(11-13): Here, the exponent -1/0.21 is employed, whereas on L.139 the exponent -4.762 is used. Please use the same notation.*
- *L.179: Insert space after "e.g.".*
- *L.181, 191: Replace closing quotation mark before "corrected" with opening quotation mark.*
- *L.186: Replace the full stop at the end of the sentence with a colon.*
- *L.190, "Expected mean drop size": Either use "expected" or "mean".*
- *L.198: Replace the reference to Uijlenhoet and Stricker (1999) by a reference to Uijlenhoet et al. (1999).*
- *L.205: "(eg. (Feller, 2010))" --> "(e.g. Feller, 2010)".*
- *L.205: Replace the full stop at the end of the sentence with a colon.*
- *L.207: "Where" --> "where".*
- *L.212: Replace the full stop at the end of the sentence with a colon.*
- *L.217: "homogeneous" --> "homogeneity".*
- *L.225: Insert "interarrival" before "time".*
- *L.238: Insert a comma before "whereas".*
- *L.239-240: Put parentheses only around publication years, i.e. use \citet instead of \citep for references in LaTeX.*
- *L.249: L.198: Replace the reference to Uijlenhoet and Stricker (1999) by a reference to Uijlenhoet et al. (1999).*
- *L.272: "Rates" --> "rates".*
- *L.274, caption Fig.4: Replace closing quotation mark before "online" with opening quotation mark (similar for "Exp" and "Corr").*
- *Caption Fig.3: "an example" --> "a sample".*
- *Fig.9: Y-axis labels of top and middle panels should also read "Mean Drop Diameter [mm]", because they represent mean diameters for each 10 s drop count.*
- *L.377: "over estimated" --> "overestimated".*
- *L.378: Insert comma before "which".*
- *L.378: "over-estimated" --> "overestimated".*
- *L.393: Insert comma before "which".*
- *L.395: Insert hyphen between "disdrometer" and "observed".*
- *L.426: "aim" --> "aims".*
- *L.443: Insert comma before "which".*
- *L.447: Insert comma before "such".*
- *L.448: Insert comma after "However".*
- *L.452: Insert comma before "making".*
- *L.460, "E.g.", "Or whether": This should be one sentence, replacing the full stop after "rainfall conditions" with a comma.*
- *L.462: "ground based" --> "ground-based".*
- *L.463: Insert comma before "respectively".*
- *L.471: Insert comma before "thus".*

- *L.478: Insert comma after "(1998)".*
- *L.482: Replace full stop at end of sentence with colon.*
- *L.492: Insert comma after "patches".*
- *L.509: "ground based" --> "ground-based".*
- *L.512: "wide scale" --> "wide-scale".*
- *L.528: In the reference list, the titles of books should be written with all nouns capitalized; the titles of papers only with the first word of the title capitalized (except for proper nouns).*

- *L.529-532: The first three references in the reference list have no author and do not appear in alphabetical order. Please correct.*
- *L.558-560: The correct journal name, issue number, page numbers and doi for this reference are: Hydrol. Earth Syst. Sci., **23**, 4737–4761, doi:10.5194/hess-23-4737-2019.*
- *L.610: Write "Poisson" with a capital "P".*

*References (if not yet listed in the manuscript)*

- *Humphrey, M.D., J.D. Istok, J.Y. Lee, J.A. Hevesi, and A.L. Flint, 1997: A new method for automated dynamic calibration of tipping-bucket rain gauges. J. Atmos. Oceanic Technol., **14**, 1513–1519, doi:10.1175/1520-0426(1997)014<1513:ANMFAD>2.0.CO;2.*
- *Uijlenhoet, R., 2001: Raindrop size distributions and radar reflectivity–rain rate relationships for radar hydrology. Hydrol. Earth Syst. Sci., **5**, 615–627, doi:10.5194/hess-5-615-2001.*
- *Uijlenhoet, R., J.M. Porrà, D. Sempere-Torres, and J.-D. Creutin, 2006: Analytical solutions to sampling effects in drop size distribution measurements during stationary rainfall: Estimation of bulk rainfall variables. J. Hydrol., **328**, 65–82, doi:10.1016/j.jhydrol.2005.11.043.*
- *Uijlenhoet, R., J.A. Smith and M. Steiner, 2003: The microphysical structure of extreme precipitation as inferred from ground-based raindrop spectra. J. Atmos. Sci., **60**, 1220–1238, doi:10.1175/1520-0469(2003)60<1220:TMSOEP>2.0.CO;2.*